# Andreev bound states at boundaries of polarized 2D Fermi superfluids with s-wave pairing and spin-orbit coupling

Kadin Thompson[1], Joachim Brand[2] and Ulrich Zuelicke[1*]

**1** Dodd-Walls Centre for Photonic and Quantum Technologies,
School of Chemical and Physical Sciences,
Victoria University of Wellington, PO Box 600,
Wellington 6140, New Zealand
**2** Dodd-Walls Centre for Photonic and Quantum Technologies,
Centre for Theoretical Chemistry and Physics,
New Zealand Institute for Advanced Study,
Massey University, Private Bag 102904,
North Shore, Auckland 0745, New Zealand

* uli.zuelicke@vuw.ac.nz

## Abstract

A topological superfluid phase characterized by an emergent chiral-$p$-wave pair potential is expected to form in a two-dimensional Fermi superfluid subject to $s$-wave pairing, spin-orbit coupling and a large-enough Zeeman splitting. Andreev bound states appear at phase boundaries, including Majorana zero modes whose existence is assured by the bulk-boundary correspondence principle. Here we study the physical properties of these subgap-energy bound states at step-like interfaces using the spin-resolved Bogoliubov–de Gennes mean-field formalism and assuming small spin-orbit coupling. Extending a recently developed spin-projection technique based on Feshbach partitioning [SciPost Phys. 5, 016 (2018)] combined with the Andreev approximation allows us to obtain remarkably simple analytical expressions for the bound-state energies as well as the majority and minority spin components of their wave functions. Besides the vacuum boundary, where a majority-spin Majorana excitation is encountered, we also consider the boundary between the topological and a nontopological superfluid phase that can appear in a coexistence scenario due to the first-order topological phase transition predicted for this system. At this superfluid-superfluid interface, we find a localized chiral Majorana mode hosted by the minority-spin sector. Our theory further predicts majority-spin subgap-energy bound states similar to those found at a Josephson junction between same-chirality $p$-wave superfluids. Their presence affects the Majorana mode due to a coupling of minority and majority spin sectors only in the small energy range where their spectra overlap. Our results may inform experimental efforts aimed at realizing and characterizing unconventional Majorana quasiparticles.





# 1 Introduction

The prospect of using condensed-matter realizations [1,2] of Majorana fermions [3,4] for fault-tolerant quantum computation [5,6] has spurred intense efforts aimed at creating topological superfluids [7–9]. One of the promising proposals [10] is based on driving two-dimensional (2D) Fermi systems with attractive *s*-wave interaction [11] and spin-orbit coupling [12,13] into a topological-superfluid (TSF) phase by increasing the Zeeman spin-splitting energy $h$ above a critical value, which is given by [14–16]

$$h_\mathrm{c} = \sqrt{\mu^2 + |\Delta|^2}, \tag{1}$$

in terms of the superfluid's chemical potential $\mu$ and *s*-wave pair-potential magnitude $|\Delta|$. In the TSF regime, the majority-spin (here: spin-↑) degrees of freedom govern the system's low-energy properties, exhibiting characteristics of a spinless chiral 2D *p*-wave superfluid [17] that is known to have topologically protected zero-energy excitations in vortices [18–22] and at its boundary [23–29]. However, the necessarily incomplete quenching of the minority-spin (i.e., spin-↓) sector can affect the system's physical properties in such a way as to spoil the perfect congruence with an ideal chiral-*p*-wave superfluid [30,31]. Understanding the effect this has on the microscopic properties of low-energy excitations emerging at the edge of a 2D TSF is one of the main purposes of the present work. The results we obtain here illuminate and extend insights gained from previous numerical studies [32,33] and related earlier work [34].

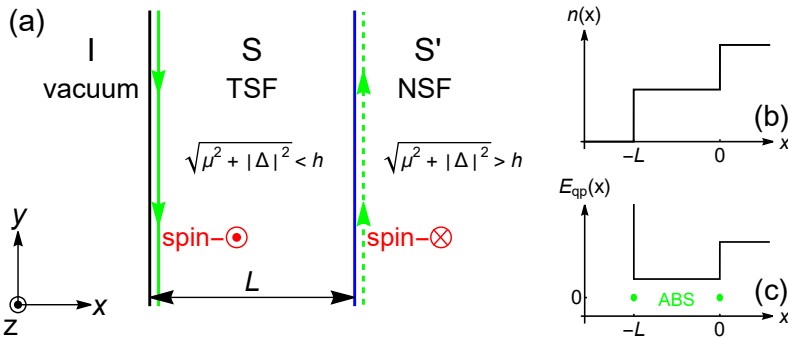

Figure 1: (a) Schematic illustration of the ISS′ model system focused on in this work. The solid and dashed green lines indicate the chiral Majorana edge modes arising at the boundary of the topological superfluid (TSF) with vacuum (I) and the nontopological superfluid (NSF), respectively, for which both spin and propagation direction turn out to be opposite. (b) Density profile for a typical physical realization of the considered ISS′ system. A 2D Fermi gas undergoing a first-order topological transition [37] splits up into coexisting TSF and NSF parts having the same chemical potential but different densities and pair-potential magnitudes. The TSF region has the smaller density and smaller pair-potential magnitude [37]. (c) The spatial variation of the minimum energy $E_{qp}$ required for quasiparticle excitations is shown as the black solid curve. We assume $E_{qp} = \infty$ in the I region so that no quasiparticles are present there. Green dots indicate the minimum energy $E_{qp} = 0$ of the interface-localized Andreev bound states (ABS) that constitute the Majorana edge modes.

Furthermore, the possibility to have situations where the system splits into coexisting TSF and nontopological-superfluid (NSF) parts [35–37] motivates our study of subgap excitations localized at a TSF-NSF interface.

The basic setup of our system of interest is shown schematically in Fig. 1(a). It constitutes an ISS′ hybrid system, where I stands for an insulator — a region of space with practically unreachable high (quasi)particle excitation energies—, while S and S′ are adjacent regions of space in which a 2D Fermi superfluid subject to $s$-wave pairing, spin-orbit coupling and Zeeman spin splitting is present in the TSF and NSF phase, respectively. To be specific, we assume that the two-particle attraction, spin-orbit coupling and Zeeman splitting are uniform across all these regions, and that the chemical potential is also the same throughout. In contrast, the $s$-wave superfluid pair potential $\Delta(\mathbf{r}) \equiv \Delta(x)$ is considered to be a function of the coordinate $x$ in the direction perpendicular to the SS′ interface. Thus, while the IS boundary arises from an ordinary single-particle confining potential, the SS′ interface is due to different superfluid order parameters being present in the S and S′ regions. In this, our envisioned realization of the TFS-NSF hybrid system differs from the one focused on in a previous study [38] where the pair-potential magnitude was assumed to be the same in the S and S′ regions but their chemical potentials were different. The intrinsically broken time-reversal symmetry due to the finite Zeeman energy distinguishes our system of interest also from the one considered in Ref. [39] where the TSF is a time-reversal invariant topological superfluid (TRITOPS).

The type of ISS′ system focused on in the present work can be realized experimentally, e.g., by a Fermi system that has split into coexisting TSF and NSF phases. Such a phase coexistence was suggested to occur in trapped 2D Fermi gases [35,36] where the external potential causes spatially varying $\mu$ and $\Delta$. As a result, $h_c$ also becomes position-dependent, making it possible for TSF and NSF regions to exist simultaneously within a trap at fixed Zeeman energy $h$. Alternatively, TSF-NSF phase coexistence arises in a 2D Fermi gas undergoing a first-order topological transition [37]. In this latter scenario, which most closely resembles the situation

investigated in the following, thermodynamics dictates the homogeneous-density values in the adjacent TSF and NSF regions and, thus, their relative size. The fact that the TSF phase generically has lower density [37] implies that it will be positioned near the system boundary and, as a result, the typical density profile of the ISS′ system has the form sketched in Fig. 1(b). To ensure that the superfluid phases in the S and S′ regions are on the BCS side of the BCS-BEC crossover [40], $\mu > 0$ is assumed throughout this work.

Obtaining the fully self-consistent spatial profile of the pair potential in a general ISS′ system will usually require numerical calculations as described, e.g., in Ref. [41]. Regardless of any details, however, it can be expected that the homogeneous-system forms of pair potentials in the individual S and S′ phases are recovered sufficiently far away from their boundaries and the SS′ interface. We therefore assume the width $L$ of the S region to be large [see illustration in Fig. 1(a)] and make the *Ansatz*

$$\Delta(x) = \left|\Delta^{(S)}\right| e^{i\varphi^{(S)}} \Theta(x + L)\Theta(-x) + \left|\Delta^{(S')}\right| e^{i\varphi^{(S')}} \Theta(x). \tag{2}$$

Here $\Theta(\cdot)$ denotes the Heaviside step function, $\varphi^{(R)} \equiv \arg\left(\Delta^{(R)}\right)$ is the phase of the complex *s*-wave pair potential in region $R \in \{S, S'\}$, and we assumed without loss of generality that the SS′ interface is located at $x = 0$. The fact that the S region is a TSF and the S′ region is a NSF, respectively, implies the relation

$$\sqrt{\mu^2 + \left|\Delta^{(S)}\right|^2} < h < \sqrt{\mu^2 + \left|\Delta^{(S')}\right|^2}, \tag{3}$$

which requires $\mu < h$ and $\left|\Delta^{(S)}\right| < \left|\Delta^{(S')}\right|$. To provide a measure for how deep into the TSF (the NSF) phase the superfluid occupying the S (the S′) region is, we introduce

$$\Delta_c = \sqrt{h^2 - \mu^2} \tag{4}$$

as the critical *s*-wave pair-potential magnitude for the hybrid system and consider situations where $\Delta_c - \left|\Delta^{(S)}\right| \gtrsim \Delta_c$ (where $\left|\Delta^{(S')}\right| - \Delta_c \sim \Delta_c$) to be well-developed TSFs (NSFs).

Due to the influence of spin-orbit coupling and Zeeman spin splitting, the minimum quasiparticle-excitation energy $E_{qp}$ is not equal to the modulus $|\Delta|$ of the *s*-wave pair potential in each respective region. A more detailed discussion of how $E_{qp}$ depends on system parameters is provided in Sec. 2 below [see Eq. (20) and following paragraph]. For our purposes, we limit consideration to the case illustrated by Fig. 1(c) where $E_{qp}^{(S)} < E_{qp}^{(S')}$, as this corresponds to the situation encountered typically in TSF-NSF hybrid systems arising from phase coexistence during a first-order topological transition [37]. Our study focuses on Andreev bound states that are localized at the individual interfaces and whose energies therefore satisfy $|E| < E_{qp}^{(S)}$. The S region also hosts extended Andreev bound states with $E_{qp}^{(S)} < |E| < E_{qp}^{(S')}$ that are confined by both interfaces and whose energies depend on $L$, but these are not further investigated here. The evanescent nature of interface-localized Andreev bound states makes it possible for the IS and SS′ interfaces to be considered independently of each other, as long as their distance $L$ is much larger than the spatial decay length for the evanescent-quasiparticle amplitudes [42]. We assume this to be the case in the following.

This article is organized as follows. The basic mathematical formalism for obtaining evanescent Andreev bound states in inhomogeneous 2D Fermi superfluids with spin-orbit coupling and Zeeman spin splitting is developed in Sec. 2. We first introduce the spin-resolved version of Bogoliubov-de Gennes theory on which our approach is based. Following that, the Feshbach-partitioning technique used previously [30] for accurate calculation of quasiparticle dispersions and associated thermodynamic properties of TSF and NSF phases is adapted and extended to obtain Andreev bound states. In particular, we show in Sec 2.1 how the excitations of the homogeneous superfluid can be faithfully represented in terms of quasiparticles

of chiral-*p*-wave superfluids associated with a fixed spin projection. The form of the specific *Ansätze* and matching conditions used for treating the IS and SS′ interfaces are then given in Secs. 2.2 and 2.3, respectively. To enable a physically realistic description of these hybrid systems, our theory accounts for the residual coupling between the effective chiral-*p*-wave superfluids realized within opposite-spin subsectors. Results obtained when applying this formalism to the edge of an individual TSF or NSF are presented in Sec 3, and the Andreev bound states localized at the TSF-NSF interface are discussed in Sec. 4. In both these cases, we obtain analytical results and use these to juxtapose the physical properties of Andreev bound states in our system of interest with those found previously for simpler systems. Our conclusions, together with a discussion and outlook on experimental ramifications of our results, are given in the final section 5. As the subgap excitations at Josephson junctions between spinless chiral-*p*-wave superfluids provide an instructive reference point to discuss our predictions, we present relevant background information in Appendix A. In the process, we also generalize previous results [43–46] to the situation where the chiral-*p*-wave pair-potential magnitudes are different on opposite sides of the junction.[1] A brief pedagogical introduction discussing the emergence and properties of Majorana edge modes at the boundary of chiral-*p*-wave superfluids is provided in Appendix B. Readers who already possess some relevant background knowledge and prefer learning first about our predictions rather than the formalism could skip parts or all of Sec. 2 on their first reading.

## 2 Formal description of interface-localized Andreev bound states

Quasiparticle states and excitation energies of a 2D Fermi superfluid with *s*-wave pairing, spin-orbit coupling and Zeeman spin splitting are obtained as solutions of the spin-resolved Bogoliubov–de Gennes (BdG) equation [50, 51]

$$
\begin{pmatrix}
\epsilon_{\hat{\mathbf{k}}} - h - \mu & 0 & \boldsymbol{\lambda} \cdot \hat{\mathbf{k}} & -\Delta(\mathbf{r}) \\
0 & -\epsilon_{\hat{\mathbf{k}}} + h + \mu & \Delta^*(\mathbf{r}) & \boldsymbol{\lambda}^* \cdot \hat{\mathbf{k}} \\
\boldsymbol{\lambda}^* \cdot \hat{\mathbf{k}} & \Delta(\mathbf{r}) & \epsilon_{\hat{\mathbf{k}}} + h - \mu & 0 \\
-\Delta^*(\mathbf{r}) & \boldsymbol{\lambda} \cdot \hat{\mathbf{k}} & 0 & -\epsilon_{\hat{\mathbf{k}}} - h + \mu
\end{pmatrix}
\begin{pmatrix}
u_\uparrow(\mathbf{r}) \\
v_\uparrow(\mathbf{r}) \\
u_\downarrow(\mathbf{r}) \\
v_\downarrow(\mathbf{r})
\end{pmatrix}
= E
\begin{pmatrix}
u_\uparrow(\mathbf{r}) \\
v_\uparrow(\mathbf{r}) \\
u_\downarrow(\mathbf{r}) \\
v_\downarrow(\mathbf{r})
\end{pmatrix}, \quad (5)
$$

which contains the *s*-wave pair potential $\Delta(\mathbf{r})$, chemical potential $\mu$ and Zeeman energy $h$ that have already been introduced above. Furthermore, '*' denotes complex conjugation, $\hat{\mathbf{k}} \equiv (-i\partial_x, -i\partial_y)$ is the 2D wave vector in position-space representation, $\epsilon_{\hat{\mathbf{k}}} \equiv \hbar^2(\hat{k}_x^2 + \hat{k}_y^2)/(2m)$ corresponds to the single-particle-energy dispersion (assumed to be parabolic for simplicity), and spin-orbit coupling is embodied in the vector $\boldsymbol{\lambda} = \lambda(i, 1)$ whose particular form represents Rashba spin-orbit coupling [52–54]. Without loss of generality, we assume $h > 0$ and $\lambda > 0$ throughout this work. Considering a piecewise-constant pair potential $\Delta(\mathbf{r}) \equiv \Delta(x)$ as per Eq. (2), a general solution of the BdG equation (5) for the entire ISS′ hybrid system is of the form

$$
\begin{pmatrix}
u_\uparrow(\mathbf{r}) \\
v_\uparrow(\mathbf{r}) \\
u_\downarrow(\mathbf{r}) \\
v_\downarrow(\mathbf{r})
\end{pmatrix}
=
\left[
\begin{pmatrix}
u_\uparrow^{(I)}(x) \\
v_\uparrow^{(I)}(x) \\
u_\downarrow^{(I)}(x) \\
v_\downarrow^{(I)}(x)
\end{pmatrix}
\Theta(-x-L) +
\begin{pmatrix}
u_\uparrow^{(S)}(x) \\
v_\uparrow^{(S)}(x) \\
u_\downarrow^{(S)}(x) \\
v_\downarrow^{(S)}(x)
\end{pmatrix}
\Theta(x+L)\Theta(-x) +
\begin{pmatrix}
u_\uparrow^{(S')}(x) \\
v_\uparrow^{(S')}(x) \\
u_\downarrow^{(S')}(x) \\
v_\downarrow^{(S')}(x)
\end{pmatrix}
\Theta(x)
\right]
e^{ik_y y}, \quad (6)
$$

---

[1]The effect of unequal pair-potential magnitudes on Andreev-bound-state spectra is implicit in theoretical treatments of *d*-wave Josephson junctions [47,48] and has also been considered for junctions of *s*-wave superfluids [49].

which is a combination of Nambu spinors that are solutions of (5) within the individual I, S, and S' regions for fixed energy $E$ and wave-vector component $k_y$, joined smoothly across the two interfaces. Based on our assumption that the 2D Fermi superfluid is homogeneous within a given region $R \in \{I, S, S'\}$, we can write the $x$-dependent part of the Nambu four-spinor pertaining to $R$ as a superposition of plane-wave eigenstates of the $4 \times 4$ BdG equation (5) with constant $\Delta(\mathbf{r}) \equiv \Delta^{(R)}$ [see Eq. (2): $\Delta^{(I)} = 0$, $\Delta^{(S)} = \left|\Delta^{(S)}\right| \exp\left(i\varphi^{(S)}\right)$ and $\Delta^{(S')} = \left|\Delta^{(S')}\right| \exp\left(i\varphi^{(S')}\right)$];

$$
\begin{pmatrix} u_\uparrow^{(R)}(x) \\ v_\uparrow^{(R)}(x) \\ u_\downarrow^{(R)}(x) \\ v_\downarrow^{(R)}(x) \end{pmatrix} = \sum_\zeta a_\zeta^{(R)} \begin{pmatrix} u_{\uparrow \mathbf{k}_\zeta^{(R)}} \\ v_{\uparrow \mathbf{k}_\zeta^{(R)}} \\ u_{\downarrow \mathbf{k}_\zeta^{(R)}} \\ v_{\downarrow \mathbf{k}_\zeta^{(R)}} \end{pmatrix} e^{i k_\zeta^{(R)} x}. \tag{7}
$$

Here $\zeta$ labels the different wave vectors $\mathbf{k}_\zeta^{(R)} \equiv \left(k_\zeta^{(R)}, k_y\right)$ for which eigenstates of (5) having energy $E$ exist, $\left(u_{\uparrow\mathbf{k}}, v_{\uparrow\mathbf{k}}, u_{\downarrow\mathbf{k}}, v_{\downarrow\mathbf{k}}\right)^T$ is the Nambu four-spinor associated with the plane-wave eigenstate of (5) with wave vector $\mathbf{k}$ and energy $E$ [see Eq. (8) below for a more explicit form of the full eigenstate], and the $a_\zeta^{(R)}$ are complex coefficients that are fixed by the requirement to satisfy the boundary conditions for region $R$.

In principle, the energy dispersions and associated eigenstates for a homogeneous 2D Fermi superfluid described by the $4 \times 4$ BdG equation (5) with constant $\Delta(\mathbf{r}) \equiv \Delta$ are available in closed-analytical form [31, 35]. However, the matching conditions for superpositions of these exact Nambu four-spinor solutions from each region at the two interfaces generate a complicated system of equations that is not straightforwardly tractable, even numerically. To circumvent this complexity and gain useful analytical insight, we use approximate expressions for energies and Nambu spinors arising from an accurate projection technique developed in Ref. [30]. We describe the fundamentals of this approach in the following subsection 2.1. Following that, the application of the formalism to obtain Andreev bound states at the individual IS and SS' interfaces is discussed in subsections 2.2 and 2.3, respectively.

## 2.1 Approximate spin-projected description of homogeneous-system excitations

A homogeneous polarized 2D Fermi superfluid with spin-orbit coupling is described by the $4 \times 4$ BdG equation (5) with constant $\Delta(\mathbf{r}) \equiv \Delta$. Using the plane-wave *Ansatz*

$$
\begin{pmatrix} u_\uparrow(\mathbf{r}) \\ v_\uparrow(\mathbf{r}) \\ u_\downarrow(\mathbf{r}) \\ v_\downarrow(\mathbf{r}) \end{pmatrix} = \begin{pmatrix} u_{\uparrow\mathbf{k}} \\ v_{\uparrow\mathbf{k}} \\ u_{\downarrow\mathbf{k}} \\ v_{\downarrow\mathbf{k}} \end{pmatrix} e^{i \mathbf{k}\cdot\mathbf{r}}, \tag{8}
$$

with 2D wave vector $\mathbf{k} \equiv (k_x, k_y)$, the BdG equation (5) is straightforwardly solved. In particular, the homogeneous-system quasiparticle-energy spectrum is found to have four dispersion branches [35, 55];

$$
E_{\mathbf{k}<\eta} = \eta \sqrt{(\epsilon_\mathbf{k} - \mu)^2 + |\Delta|^2 + h^2 + \lambda^2 \mathbf{k}^2 - 2\sqrt{(\epsilon_\mathbf{k} - \mu)^2 (h^2 + \lambda^2 \mathbf{k}^2) + |\Delta|^2 h^2}}, \tag{9a}
$$

$$
E_{\mathbf{k}>\eta} = \eta \sqrt{(\epsilon_\mathbf{k} - \mu)^2 + |\Delta|^2 + h^2 + \lambda^2 \mathbf{k}^2 + 2\sqrt{(\epsilon_\mathbf{k} - \mu)^2 (h^2 + \lambda^2 \mathbf{k}^2) + |\Delta|^2 h^2}}, \tag{9b}
$$

with $\eta \in \{+, -\}$ distinguishing positive-energy and negative-energy states.

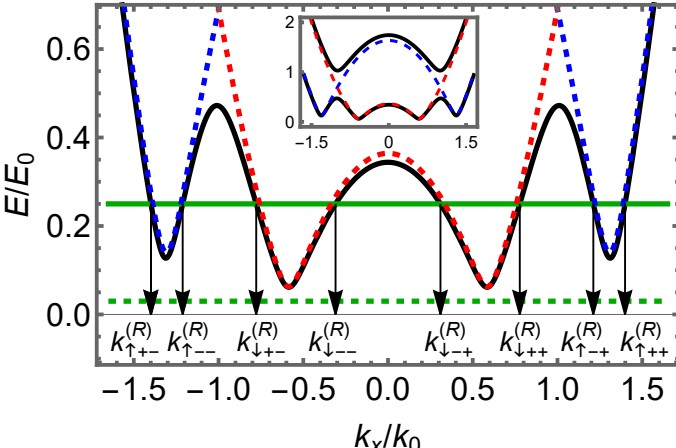

Figure 2: Approximate description of the low-energy quasiparticle dispersion in the nontopological-superfluid (NSF) phase. The black solid curve shows the lower positive-energy branch $E_{\mathbf{k}<+}$ [see explicit expression given in Eq. (9a)] of quasiparticle energies obtained from diagonalizing the Hamiltonian in Eq. (5) for $k_y = 0$ and with the parameters $\mu/E_0 = 1.00$, $|\Delta|/E_0 = 0.30$, $h/E_0 = 0.70$, $\lambda k_0/E_0 = 0.25$. Here $k_0$ and $E_0$ are arbitrary wave-number and energy units related via $E_0 = \hbar^2 k_0^2/(2m)$. For reference, the inset plots both positive-energy branches of the quasiparticle-excitation spectrum [$E_{\mathbf{k}<+}$ and $E_{\mathbf{k}>+}$ as per Eqs. (9)] as black solid curves. The blue dashed (red dashed) curve is the approximate energy dispersion $E_{\mathbf{k}\uparrow+}$ ($E_{\mathbf{k}\downarrow+}$) from Eq. (18). Intersection of the dispersions with a fixed-energy value $E$ above the quasiparticle-excitation gap (solid green horizontal line) defines wave-vector components $k_{\sigma\tau\alpha}^{(R)}$ associated with quasiparticle excitations moving perpendicular to the interfaces in region $R$, for which approximate analytical expressions are given in Eq. (22). For a value of $E$ below the quasiparticle-excitation gap (dashed green horizontal line), these wave numbers are complex-valued.

### 2.1.1 Formally exact 2×2 projection

The inset of Fig. 2 shows a plot of the positive-energy branches $E_{\mathbf{k}<+}$ and $E_{\mathbf{k}>+}$. It is possible to represent the low-energy part of these dispersions quite faithfully in terms of a set of spin-projected dispersion relations (shown as blue and red dashed curves in Fig. 2). This approach is motivated by the observation that, for small-enough pair-potential magnitude and not-too-large spin-orbit-coupling strength, as compared with the spin-polarizing Zeeman energy,[2] the plane-wave Nambu eigenspinors (8) that are the solutions of (5) for the homogeneous system have generally either dominating spin-$\uparrow$ character or dominating spin-$\downarrow$ character, i.e., they satisfy

$$|u_{\sigma\mathbf{k}}|^2 + |v_{\sigma\mathbf{k}}|^2 \gg |u_{\bar{\sigma}\mathbf{k}}|^2 + |v_{\bar{\sigma}\mathbf{k}}|^2 \,, \tag{10}$$

with $\bar{\sigma}$ being the opposite of $\sigma$ [i.e., $\bar{\sigma} = \downarrow$ ($\uparrow$) if $\sigma = \uparrow$ ($\downarrow$)]. The formal rewriting of the spin-resolved BdG equation (5) for constant $\Delta(\mathbf{r}) \equiv \Delta$ in plane-wave representation in terms

---

[2]Specifically, $|\Delta|^2 \ll \mu h$ and $\lambda^2 \ll \hbar^2 h/m$ are the required conditions. The first of these typically applies in the BCS regime for $s$-wave pairing, but the approach turns out to also describe the TSF in the BEC regime [31].

of the two $2 \times 2$ equations

$$\left(\mathcal{H}_{\sigma\sigma} - \mathcal{H}_{\sigma\bar{\sigma}} \left[\mathcal{H}_{\bar{\sigma}\bar{\sigma}} - E \, \mathbb{1}\right]^{-1} \mathcal{H}_{\bar{\sigma}\sigma}\right) \begin{pmatrix} u_{\sigma\mathbf{k}} \\ v_{\sigma\mathbf{k}} \end{pmatrix} = E \begin{pmatrix} u_{\sigma\mathbf{k}} \\ v_{\sigma\mathbf{k}} \end{pmatrix}, \tag{11a}$$

$$\begin{pmatrix} u_{\bar{\sigma}\mathbf{k}} \\ v_{\bar{\sigma}\mathbf{k}} \end{pmatrix} = -\left[\mathcal{H}_{\bar{\sigma}\bar{\sigma}} - E \, \mathbb{1}\right]^{-1} \mathcal{H}_{\bar{\sigma}\sigma} \begin{pmatrix} u_{\sigma\mathbf{k}} \\ v_{\sigma\mathbf{k}} \end{pmatrix}, \tag{11b}$$

where

$$\mathcal{H}_{\uparrow\uparrow} = \begin{pmatrix} \epsilon_{\mathbf{k}} - h - \mu & 0 \\ 0 & -\epsilon_{\mathbf{k}} + h + \mu \end{pmatrix}, \quad \mathcal{H}_{\downarrow\downarrow} = \begin{pmatrix} \epsilon_{\mathbf{k}} + h - \mu & 0 \\ 0 & -\epsilon_{\mathbf{k}} - h + \mu \end{pmatrix}, \tag{12a}$$

$$\mathcal{H}_{\uparrow\downarrow} \equiv \left(\mathcal{H}_{\downarrow\uparrow}\right)^{\dagger} = \begin{pmatrix} \boldsymbol{\lambda} \cdot \mathbf{k} & -\Delta \\ \Delta^* & \boldsymbol{\lambda}^* \cdot \mathbf{k} \end{pmatrix}, \tag{12b}$$

forms the basis for an approximate treatment where Nambu four-spinor solutions of (5) for the homogeneous system are represented in terms of their respective large amplitudes $u_{\sigma\mathbf{k}}$ and $v_{\sigma\mathbf{k}}$. Specializing the more general expressions from Ref. [30] to the case of weak spin-orbit coupling $\lambda|\mathbf{k}| \ll h$, we find

$$\begin{pmatrix} \xi_{\mathbf{k}\uparrow} & -\frac{\boldsymbol{\lambda}\cdot\mathbf{k}}{h}\Delta \\ -\frac{\boldsymbol{\lambda}^*\cdot\mathbf{k}}{h}\Delta^* & -\xi_{\mathbf{k}\uparrow} \end{pmatrix} \begin{pmatrix} u_{\uparrow\mathbf{k}} \\ v_{\uparrow\mathbf{k}} \end{pmatrix} = E_{\mathbf{k}\uparrow} \begin{pmatrix} u_{\uparrow\mathbf{k}} \\ v_{\uparrow\mathbf{k}} \end{pmatrix}, \tag{13a}$$

$$\begin{pmatrix} u_{\downarrow\mathbf{k}} \\ v_{\downarrow\mathbf{k}} \end{pmatrix} = \begin{pmatrix} -\frac{\boldsymbol{\lambda}^*\cdot\mathbf{k}}{2h} & -\frac{\Delta}{2h} \\ -\frac{\Delta^*}{2h} & \frac{\boldsymbol{\lambda}\cdot\mathbf{k}}{2h} \end{pmatrix} \begin{pmatrix} u_{\uparrow\mathbf{k}} \\ v_{\uparrow\mathbf{k}} \end{pmatrix}, \tag{13b}$$

for the large-spin-$\uparrow$ spinors, and similarly

$$\begin{pmatrix} \xi_{\mathbf{k}\downarrow} & -\frac{\boldsymbol{\lambda}^*\cdot\mathbf{k}}{h}\Delta \\ -\frac{\boldsymbol{\lambda}\cdot\mathbf{k}}{h}\Delta^* & -\xi_{\mathbf{k}\downarrow} \end{pmatrix} \begin{pmatrix} u_{\downarrow\mathbf{k}} \\ v_{\downarrow\mathbf{k}} \end{pmatrix} = E_{\mathbf{k}\downarrow} \begin{pmatrix} u_{\downarrow\mathbf{k}} \\ v_{\downarrow\mathbf{k}} \end{pmatrix}, \tag{14a}$$

$$\begin{pmatrix} u_{\uparrow\mathbf{k}} \\ v_{\uparrow\mathbf{k}} \end{pmatrix} = \begin{pmatrix} \frac{\boldsymbol{\lambda}\cdot\mathbf{k}}{2h} & -\frac{\Delta}{2h} \\ -\frac{\Delta^*}{2h} & -\frac{\boldsymbol{\lambda}^*\cdot\mathbf{k}}{2h} \end{pmatrix} \begin{pmatrix} u_{\downarrow\mathbf{k}} \\ v_{\downarrow\mathbf{k}} \end{pmatrix}, \tag{14b}$$

for the large-spin-$\downarrow$ spinors. Here we introduced the effective unpaired-quasiparticle energy

$$\xi_{\mathbf{k}\sigma} = \frac{\hbar^2}{2m_{\sigma}}\mathbf{k}^2 + \nu_{\sigma} - \mu. \tag{15}$$

Within our leading-order-in-$\lambda|\mathbf{k}|/h$ approximation, which amounts to keeping terms upto quadratic order in $\lambda|\mathbf{k}|/h$ in the effective $2 \times 2$ BdG Hamiltonians of Eqs. (13a) and (14a) but only terms upto linear order in $\lambda|\mathbf{k}|/h$ in the determining relations (13b) and (14b) for the small spinor amplitudes, we have

$$m_{\uparrow} = \frac{m}{1 - \frac{m\lambda^2}{\hbar^2 h}\left(1 + \frac{|\Delta|^2}{4h^2}\right)}, \qquad m_{\downarrow} = \frac{m}{1 + \frac{m\lambda^2}{\hbar^2 h}\left(1 + \frac{|\Delta|^2}{4h^2}\right)}, \tag{16}$$

for the spin-dependent effective quasiparticle mass, and

$$\nu_{\uparrow} = -h + \frac{|\Delta|^2}{2h}, \qquad \nu_{\downarrow} = h - \frac{|\Delta|^2}{2h}, \tag{17}$$

for the spin-dependent band-bottom shift entering Eq. (15).

### 2.1.2 Energy dispersions for weak spin-orbit coupling

Diagonalization of (13a) and (14a) yields the dominant-spin-$\sigma$-quasiparticle energy dispersions

$$E_{\mathbf{k}\sigma\eta} = \eta \sqrt{\xi_{\mathbf{k}\sigma}^2 + \frac{\lambda^2|\Delta|^2}{h^2}\mathbf{k}^2} \equiv \eta \sqrt{\left(\xi_{\mathbf{k}\sigma} + \frac{\hbar^2}{m_\sigma}k_{\Delta\sigma}^2\right)^2 + \bar{\Delta}_\sigma^2}, \qquad (18)$$

where we introduced the abbreviations

$$\bar{\Delta}_\sigma = \Delta_\sigma \sqrt{1 - \frac{k_{\Delta\sigma}^2}{k_{\mathrm{F}\sigma}^2}}, \quad \Delta_\sigma = \frac{\lambda\, k_{\mathrm{F}\sigma}}{h}|\Delta|, \quad k_{\mathrm{F}\sigma} = \sqrt{\frac{2m_\sigma}{\hbar^2}(\mu - \nu_\sigma)}, \quad k_{\Delta\sigma} = \frac{m_\sigma}{\hbar^2}\frac{\lambda|\Delta|}{h}. \quad (19)$$

Although $|k_{\Delta\sigma}| \ll |k_{\mathrm{F}\sigma}|$ holds typically, we have retained $k_{\Delta\sigma}$-dependent terms in the above expressions for the sake of formal consistency. The dispersions (18) formally coincide with the quasiparticle energies obtained for $p$-wave pairing [56] and reproduce the low-energy part of the true quasiparticle-excitation spectrum. See Fig. 2 for a comparison. Straightforward calculation yields the minimum excitation energy of the spin-$\sigma$ branch as [57]

$$\min\left|E_{\mathbf{k}\sigma\eta}\right| = \begin{cases} |\bar{\Delta}_\sigma|, & \text{occurring where } |\mathbf{k}| = \sqrt{k_{\mathrm{F}\sigma}^2 - 2k_{\Delta\sigma}^2}, & \text{for } k_{\mathrm{F}\sigma}^2 \geq 2k_{\Delta\sigma}^2, \\ |\mu - \nu_\sigma|, & \text{occurring at } \mathbf{k} = \mathbf{0}, & \text{for } k_{\mathrm{F}\sigma}^2 < 2k_{\Delta\sigma}^2. \end{cases} \quad (20)$$

Thus the criterion $\min|E_{\mathbf{k}\downarrow\eta}| = 0$ for the topological transition [14–16] yields the condition

$$\mu = h_c - \frac{|\Delta|^2}{2h_c}, \qquad (21)$$

for the critical Zeeman energy within the approximate projected theory, consistent with the leading-order small-$(|\Delta|/h)$ expansion of the exact relation $\mu = \sqrt{h_c^2 - |\Delta|^2}$ obtained by rearranging (1) under the assumption that $\mu > 0$. Furthermore, in the NSF phase, the relations $0 < k_{\mathrm{F}\downarrow} < k_{\mathrm{F}\uparrow}$ imply $|\bar{\Delta}_\downarrow| < |\bar{\Delta}_\uparrow|$. Hence, the minimum quasiparticle-excitation energy of the NSF is $E_{\mathrm{qp}}^{\mathrm{(NSF)}} \equiv \left|\bar{\Delta}_\downarrow^{\mathrm{(NSF)}}\right|$. This is in contrast to the TSF phase where, except close to the transition, the majority-spin excitation gap $|\bar{\Delta}_\uparrow|$ is smaller than the minimum of the spin-$\downarrow$ dispersion at $\mathbf{k} = \mathbf{0}$ and, thus, $E_{\mathrm{qp}}^{\mathrm{(TSF)}} \equiv \left|\bar{\Delta}_\uparrow^{\mathrm{(TSF)}}\right|$. The respective magnitudes of $E_{\mathrm{qp}}^{\mathrm{(TSF)}}$ and $E_{\mathrm{qp}}^{\mathrm{(NSF)}}$ in a TSF-NSF hybrid system generally depend on physical details. For our present work, we envision a situation where $E_{\mathrm{qp}}^{\mathrm{(TSF)}} < E_{\mathrm{qp}}^{\mathrm{(NSF)}}$, as illustrated in Fig. 1(c).

Within our approach, the spin-labelled large components of each Nambu four-spinor are associated with the quasiparticle dynamics via the respective $2 \times 2$ BdG equations (13a) and (14a). However, the small components in the four-spinors are not discarded but, rather, determined via the relations (13b) and (14b) for inclusion in relevant fomulae for physical quantities. This formalism has been shown [30] to yield accurate results for thermodynamic properties such as the quasiparticle-density distribution and self-consistently determined chemical and pair potentials. Here we apply this method to the calculation of Andreev bound states in the ISS′ system depicted in Fig. 1 for which consideration of the entire four-spinor wave function is essential [58]. The formalism developed in the remainder of this subsection forms the basis for the theoretical description of the IS and SS′ interfaces discussed in Secs. 2.2 and 2.3 below.

### 2.1.3 Wave numbers for interface matching

We first obtain accurate approximations for the wave numbers $k_\zeta^{(R)}$ appearing in the *Ansatz* (7) by inverting the relations $E_{\mathbf{k}\sigma\eta} = E$ for fixed $E$, using the quasiparticle-energy dispersions

from Eq. (18) within a particular region $R \in \{I, S, S'\}$. This procedure yields

$$k_{\sigma\tau\alpha}^{(R)} = \alpha \sqrt{\left(k_{F\sigma}^{(R)}\right)^2 - k_y^2 - 2\left(k_{\Delta\sigma}^{(R)}\right)^2 + \tau \frac{2m_\sigma^{(R)}}{\hbar^2} \sqrt{E^2 - \left(\bar{\Delta}_\sigma^{(R)}\right)^2}}, \qquad (22)$$

where the superscript $(R)$ is now used liberally to indicate that quantities pertain to region $R$. The label $\alpha \in \{+, -\}$ distinguishes right-moving ($\alpha = +$) and left-moving ($\alpha = -$) states, while $\tau \in \{+, -\}$ labels quasiparticle ($\tau = +$) and quasihole ($\tau = -$) excitations. Figure 2 illustrates the origin of this nomenclature, including the spin-↑ (spin-↓) label for those wave numbers that arise from parts of the dispersion approximated by the blue (red) dashed curves. The Nambu two-spinor solutions of the spin-$\sigma$ $2 \times 2$ BdG equations [(13a) for spin-↑ and (14a) for spin-↓] associated with energy $E$ and 2D wave vector $\mathbf{k}_{\sigma\tau\alpha}^{(R)} = \left(k_{\sigma\tau\alpha}^{(R)}, k_y\right)$ are found as

$$u_{\sigma\tau\alpha}^{(R)} = -\varsigma i \exp\left(i\varphi^{(R)}\right) \frac{k_{\sigma\tau\alpha}^{(R)} - \varsigma i k_y}{\sqrt{\left(k_{\sigma\tau\alpha}^{(R)}\right)^2 + k_y^2}} \sqrt{\frac{E - \frac{\hbar^2}{m_\sigma^{(R)}} \left(k_{\Delta\sigma}^{(R)}\right)^2 + \tau \sqrt{E^2 - \left(\bar{\Delta}_\sigma^{(R)}\right)^2}}{2E}}, \qquad (23a)$$

$$v_{\sigma\tau\alpha}^{(R)} = \operatorname{sgn}(E) \sqrt{\frac{E + \frac{\hbar^2}{m_\sigma^{(R)}} \left(k_{\Delta\sigma}^{(R)}\right)^2 - \tau \sqrt{E^2 - \left(\bar{\Delta}_\sigma^{(R)}\right)^2}}{2E}}, \qquad (23b)$$

with $\operatorname{sgn}(\cdot)$ denoting the sign function and $\varsigma = + (-)$ for $\sigma = \uparrow (\downarrow)$. Using also the relations (13b) and (14b) for a Nambu four-spinor's small components in terms of the dominant-spin ones, it is possible to express the Feshbach-projection-approximated form of the full four-component Nambu eigenspinor of the BdG equation (5) in region $R$ with energy $E$ and wave vector $\mathbf{k}_{\sigma\tau\alpha}^{(R)}$ in terms of that eigenspinor's dominating (large-spin-$\sigma$) part $\left(u_{\sigma\tau\alpha}^{(R)}, v_{\sigma\tau\alpha}^{(R)}\right)^T$ as

$$\begin{pmatrix} u_{\uparrow\mathbf{k}_{\sigma\tau\alpha}^{(R)}} \\ v_{\uparrow\mathbf{k}_{\sigma\tau\alpha}^{(R)}} \\ u_{\downarrow\mathbf{k}_{\sigma\tau\alpha}^{(R)}} \\ v_{\downarrow\mathbf{k}_{\sigma\tau\alpha}^{(R)}} \end{pmatrix} = \mathcal{M}_{\sigma\tau\alpha}^{(R)} \begin{pmatrix} u_{\sigma\tau\alpha}^{(R)} \\ v_{\sigma\tau\alpha}^{(R)} \end{pmatrix}. \qquad (24)$$

The $4 \times 2$ matrices

$$\mathcal{M}_{\uparrow\tau\alpha}^{(R)} = \begin{pmatrix} \mathbb{1}_{2\times2} \\ \mathcal{J}_{\uparrow\tau\alpha}^{(R)} \end{pmatrix}, \quad \mathcal{M}_{\downarrow\tau\alpha}^{(R)} = \begin{pmatrix} \mathcal{J}_{\downarrow\tau\alpha}^{(R)} \\ \mathbb{1}_{2\times2} \end{pmatrix}, \qquad (25)$$

contain the $2 \times 2$ unit matrix $\mathbb{1}_{2\times2}$, and the remaining $2 \times 2$ blocks

$$\mathcal{J}_{\uparrow\tau\alpha}^{(R)} = \frac{1}{2h} \begin{pmatrix} \lambda\left(ik_{\uparrow\tau\alpha}^{(R)} - k_y\right) & -\Delta^{(R)} \\ -\left(\Delta^{(R)}\right)^* & \lambda\left(ik_{\uparrow\tau\alpha}^{(R)} + k_y\right) \end{pmatrix}, \quad \mathcal{J}_{\downarrow\tau\alpha}^{(R)} = \frac{1}{2h} \begin{pmatrix} \lambda\left(ik_{\downarrow\tau\alpha}^{(R)} + k_y\right) & -\Delta^{(R)} \\ -\left(\Delta^{(R)}\right)^* & \lambda\left(ik_{\downarrow\tau\alpha}^{(R)} - k_y\right) \end{pmatrix}, \qquad (26)$$

are obtained by specializing the matrices entering the relations (13b) and (14b) between small and large components of Nambu four-spinors to the situation where $\mathbf{k} = \mathbf{k}_{\sigma\tau\alpha}^{(R)}$.

### 2.1.4 Andreev approximation

In the following two subsections 2.2 and 2.3, the general formalism developed here will be applied to describe bound states at the IS and SS′ interfaces, respectively. Useful physical insights emerge from analytical results that are obtained using the Andreev approximation [59].

In the present context, this approach amounts to neglecting $k_{\Delta\sigma}$ and further approximating Eq. (22) by

$$k_{\sigma\tau\alpha}^{(R)} \approx \alpha\, k_{F\sigma}^{(R)} + \alpha\tau\, \frac{m_\sigma^{(R)}}{\hbar^2 k_{F\sigma}^{(R)}} \sqrt{E^2 - \left(\Delta_\sigma^{(R)}\right)^2}. \tag{27}$$

To ensure that (27) is a good approximation, $|E|, |\Delta_\sigma^{(R)}| \ll |\mu - \nu_\sigma^{(R)}|$ and $|k_y| \ll |k_{F\sigma}^{(R)}|$ are required, which also guarantee $|k_{\Delta\sigma}| \ll |k_{F\sigma}^{(R)}|$. More explicitly, the condition for the validity of the Andreev approximation is expressed as

$$1 \gg \frac{m_\sigma^{(R)}\left|\Delta_\sigma^{(R)}\right|}{\hbar^2\left|k_{F\sigma}^{(R)}\right|^2} \equiv \frac{\lambda\left|k_{F\sigma}^{(R)}\right|}{h}\, \frac{m_\sigma^{(R)}\left|\Delta^{(R)}\right|}{\hbar^2\left|k_{F\sigma}^{(R)}\right|^2} = \mathcal{O}\left(\sqrt{\frac{m\,\lambda^2}{\hbar^2 h}}\right) \cdot \mathcal{O}\left(\sqrt{\frac{m_\sigma^{(R)}\left|\Delta^{(R)}\right|^2}{\hbar^2\left|k_{F\sigma}^{(R)}\right|^2 h}}\right). \tag{28}$$

Thus, in our system of interest, applicability of the Andreev approximation can be guaranteed by the smallness of the *s*-wave pair potential compared to the Fermi energy (the usual condition [59]) and/or a small magnitude of the spin-orbit coupling. Using the Andreev approximation in the Bogoliubov-quasiparticle spinors yields

$$u_{\sigma\tau\alpha}^{(R)} \approx \begin{cases} -\varsigma\alpha\, i\, \exp\left[i\left(\varphi^{(R)} + \frac{\tau\,\mathrm{sgn}(E)}{2}\theta_\sigma^{(R)} - \varsigma\alpha\vartheta_{k_y\sigma}^{(R)}\right)\right] \sqrt{\frac{\Delta_\sigma^{(R)}}{2|E|}}, & k_{F\sigma}^{(R)} = \left|k_{F\sigma}^{(R)}\right|, \\[2ex] -\varsigma\alpha\, i\, \exp\left(i\varphi^{(R)}\right) \sqrt{\frac{\left|k_{F\sigma}^{(R)}\right| - \varsigma\alpha k_y}{\left|k_{F\sigma}^{(R)}\right| + \varsigma\alpha k_y}} \sqrt{\frac{E + \tau\sqrt{E^2 + \left|\Delta_\sigma^{(R)}\right|^2}}{2E}}, & k_{F\sigma}^{(R)} = i\left|k_{F\sigma}^{(R)}\right|, \end{cases} \tag{29a}$$

$$v_{\sigma\tau\alpha}^{(R)} \approx \begin{cases} \mathrm{sgn}(E)\, \exp\left(-i\frac{\tau\,\mathrm{sgn}(E)}{2}\theta_\sigma^{(R)}\right) \sqrt{\frac{\Delta_\sigma^{(R)}}{2|E|}}, & k_{F\sigma}^{(R)} = \left|k_{F\sigma}^{(R)}\right|, \\[2ex] \mathrm{sgn}(E)\, \sqrt{\frac{E - \tau\sqrt{E^2 + \left|\Delta_\sigma^{(R)}\right|^2}}{2E}}, & k_{F\sigma}^{(R)} = i\left|k_{F\sigma}^{(R)}\right|, \end{cases} \tag{29b}$$

with $\theta_\sigma^{(R)} = \arccos\left(|E|/\Delta_\sigma^{(R)}\right)$, $\vartheta_{k_y\sigma}^{(R)} = \arcsin\left(k_y/k_{F\sigma}^{(R)}\right)$, and again $\varsigma = +\,(-)$ for $\sigma = \uparrow\,(\downarrow)$.

## 2.2 IS interface — edge states

Our particular system of interest has an IS interface where the S region is in the TSF phase [see Fig. 1(a)]. For completeness and to provide further insight through comparisons, we consider in the following also the case of an IS interface where S is an NSF.

We assume that the I region is inaccessible to quasiparticle excitations, i.e., its presence imposes the hard-wall boundary condition

$$\begin{pmatrix} u_\uparrow^{(S)}(-L) \\ v_\uparrow^{(S)}(-L) \\ u_\downarrow^{(S)}(-L) \\ v_\downarrow^{(S)}(-L) \end{pmatrix} = \begin{pmatrix} 0 \\ 0 \\ 0 \\ 0 \end{pmatrix}. \tag{30}$$

To describe Andreev bound states localized at the IS interface via the *Ansatz* (7) with $\mathbf{k}_\zeta \to \mathbf{k}_{\sigma\tau\alpha}^{(S)} = \left(k_{\sigma\tau\alpha}^{(S)}, k_y\right)$, only evanescent states having $\Im\left(k_{\sigma\tau\alpha}^{(S)}\right) > 0$ should be included. Inspection of Eqs. (22) and/or (27) shows that, in regions where $\mu - \nu_\sigma^{(R)} > 0$ and $k_{F\sigma}^{(R)}$ is therefore real, $k_{\sigma\tau\alpha}^{(R)}$ becomes complex-valued for $|E| < \Delta_\sigma^{(R)}$, i.e., subgap-energy states. If $\mu - \nu_\sigma^{(R)} < 0$ in region $R$, $k_{F\sigma}^{(R)}$ and $k_{\sigma\tau\alpha}^{(R)}$ are both purely imaginary for subgap-energy states.

Hence, when S is in the NSF phase, states to include in the *Ansatz* (7) satisfy $|E| < E_{\text{qp}}^{(\text{NSF})} \approx \Delta_{\downarrow}^{(\text{S})}$ and $\alpha\tau = +$. In contrast, when the S region hosts a TSF, $|E| < E_{\text{qp}}^{(\text{TSF})} \approx \Delta_{\uparrow}^{(\text{S})}$ is required together with $\alpha\tau = +$ for $\sigma = \uparrow$ and $\alpha = +$ for $\sigma = \downarrow$. Based on these considerations, we find the explicit form of the Nambu spinor for S being a NSF as

$$
\begin{pmatrix} u_{\uparrow}^{(\text{NSF})}(x) \\ v_{\uparrow}^{(\text{NSF})}(x) \\ u_{\downarrow}^{(\text{NSF})}(x) \\ v_{\downarrow}^{(\text{NSF})}(x) \end{pmatrix} = \sum_{\sigma} \left[ a_{\sigma++}^{(\text{S})} \, \mathcal{M}_{\sigma++}^{(\text{S})} \begin{pmatrix} u_{\sigma++}^{(\text{S})} \\ v_{\sigma++}^{(\text{S})} \end{pmatrix} e^{i k_{\sigma++}^{(\text{S})} x} + a_{\sigma--}^{(\text{S})} \, \mathcal{M}_{\sigma--}^{(\text{S})} \begin{pmatrix} u_{\sigma--}^{(\text{S})} \\ v_{\sigma--}^{(\text{S})} \end{pmatrix} e^{i k_{\sigma--}^{(\text{S})} x} \right] , \quad (31)
$$

while the allowed superposition for the case where S is a TSF has the form

$$
\begin{pmatrix} u_{\uparrow}^{(\text{TSF})}(x) \\ v_{\uparrow}^{(\text{TSF})}(x) \\ u_{\downarrow}^{(\text{TSF})}(x) \\ v_{\downarrow}^{(\text{TSF})}(x) \end{pmatrix} = a_{\uparrow++}^{(\text{S})} \, \mathcal{M}_{\uparrow++}^{(\text{S})} \begin{pmatrix} u_{\uparrow++}^{(\text{S})} \\ v_{\uparrow++}^{(\text{S})} \end{pmatrix} e^{i k_{\uparrow++}^{(\text{S})} x} + a_{\uparrow--}^{(\text{S})} \, \mathcal{M}_{\uparrow--}^{(\text{S})} \begin{pmatrix} u_{\uparrow--}^{(\text{S})} \\ v_{\uparrow--}^{(\text{S})} \end{pmatrix} e^{i k_{\uparrow--}^{(\text{S})} x}
$$

$$
+ a_{\downarrow++}^{(\text{S})} \, \mathcal{M}_{\downarrow++}^{(\text{S})} \begin{pmatrix} u_{\downarrow++}^{(\text{S})} \\ v_{\downarrow++}^{(\text{S})} \end{pmatrix} e^{i k_{\downarrow++}^{(\text{S})} x} + a_{\downarrow-+}^{(\text{S})} \, \mathcal{M}_{\downarrow-+}^{(\text{S})} \begin{pmatrix} u_{\downarrow-+}^{(\text{S})} \\ v_{\downarrow-+}^{(\text{S})} \end{pmatrix} e^{i k_{\downarrow-+}^{(\text{S})} x} . \quad (32)
$$

Requiring (30) to hold for *Ansatz* (31) or (32), respectively, yields a homogeneous system of four linear equations for the four coefficients $a_{\sigma\tau\alpha}^{(\text{S})}$ appearing in that particular *Ansatz* whose characteristic equation determines the Andreev-edge-state energies. As the fully general form of the characteristic equations is quite long and rather unilluminating [60], we only present results here after applying further approximations.

If entries $\mathcal{O}\big(|\Delta^{(R)}|/h\big)$ in the matrices $\mathcal{J}_{\sigma\alpha}^{(R)}$ defined in Eq. (26) are neglected, we obtain the result

$$
\left\{ \left[ 1 - \frac{\lambda^2}{4h^2} i k_y \left( k_{\uparrow++}^{(\text{S})} - k_{\uparrow--}^{(\text{S})} \right) \right] u_{\uparrow++}^{(\text{S})} v_{\uparrow--}^{(\text{S})} - \left[ 1 + \frac{\lambda^2}{4h^2} i k_y \left( k_{\uparrow++}^{(\text{S})} - k_{\uparrow--}^{(\text{S})} \right) \right] u_{\uparrow--}^{(\text{S})} v_{\uparrow++}^{(\text{S})} \right\}
$$

$$
\times \left\{ \left[ 1 + \frac{\lambda^2}{4h^2} i k_y \left( k_{\downarrow++}^{(\text{S})} - k_{\downarrow--}^{(\text{S})} \right) \right] u_{\downarrow++}^{(\text{S})} v_{\downarrow--}^{(\text{S})} - \left[ 1 - \frac{\lambda^2}{4h^2} i k_y \left( k_{\downarrow++}^{(\text{S})} - k_{\downarrow--}^{(\text{S})} \right) \right] u_{\downarrow--}^{(\text{S})} v_{\downarrow++}^{(\text{S})} \right\}
$$

$$
= \frac{\lambda^2}{4h^2} \left( k_{\uparrow++}^{(\text{S})} - k_{\uparrow--}^{(\text{S})} \right) \left( k_{\downarrow++}^{(\text{S})} - k_{\downarrow--}^{(\text{S})} \right) \left( u_{\uparrow++}^{(\text{S})} v_{\downarrow++}^{(\text{S})} - u_{\downarrow++}^{(\text{S})} v_{\uparrow++}^{(\text{S})} \right) \left( u_{\uparrow--}^{(\text{S})} v_{\downarrow--}^{(\text{S})} - u_{\downarrow--}^{(\text{S})} v_{\uparrow--}^{(\text{S})} \right) , \quad (33)
$$

for the characteristic equation in the NSF case. As the *Ansätze* Eqs. (31) and (32) for the I-NSF and I-TSF boundaries differ only by the replacement of the state with label $\sigma\tau\alpha \equiv \downarrow--$ in the former by the state labelled $\sigma\tau\alpha \equiv \downarrow-+$ in the latter, the characteristic equation for the situation where S is in the TSF phase can be obtained by making the corresponding adjustments in Eq. (33), yielding

$$
\left\{ \left[ 1 - \frac{\lambda^2}{4h^2} i k_y \left( k_{\uparrow++}^{(\text{S})} - k_{\uparrow--}^{(\text{S})} \right) \right] u_{\uparrow++}^{(\text{S})} v_{\uparrow--}^{(\text{S})} - \left[ 1 + \frac{\lambda^2}{4h^2} i k_y \left( k_{\uparrow++}^{(\text{S})} - k_{\uparrow--}^{(\text{S})} \right) \right] u_{\uparrow--}^{(\text{S})} v_{\uparrow++}^{(\text{S})} \right\}
$$

$$
\times \left\{ \left[ 1 + \frac{\lambda^2}{4h^2} i k_y \left( k_{\downarrow++}^{(\text{S})} - k_{\downarrow-+}^{(\text{S})} \right) \right] u_{\downarrow++}^{(\text{S})} v_{\downarrow-+}^{(\text{S})} - \left[ 1 - \frac{\lambda^2}{4h^2} i k_y \left( k_{\downarrow++}^{(\text{S})} - k_{\downarrow-+}^{(\text{S})} \right) \right] u_{\downarrow-+}^{(\text{S})} v_{\downarrow++}^{(\text{S})} \right\}
$$

$$
= \frac{\lambda^2}{4h^2} \left( k_{\uparrow++}^{(\text{S})} - k_{\uparrow--}^{(\text{S})} \right) \left( k_{\downarrow++}^{(\text{S})} - k_{\downarrow-+}^{(\text{S})} \right) \left( u_{\uparrow++}^{(\text{S})} v_{\downarrow++}^{(\text{S})} - u_{\downarrow++}^{(\text{S})} v_{\uparrow++}^{(\text{S})} \right) \left( u_{\uparrow--}^{(\text{S})} v_{\downarrow-+}^{(\text{S})} - u_{\downarrow-+}^{(\text{S})} v_{\uparrow--}^{(\text{S})} \right) . \quad (34)
$$

Application of the Andreev-approximation expressions from Eqs. (27) and (29) transforms the characteristic equation (33) for the I-NSF system into the form

$$
\cos\left[\theta_\uparrow^{(S)} - \mathrm{sgn}(E)\left(\vartheta_{k_y\uparrow}^{(S)} + \frac{\lambda^2}{2h^2}k_{F\uparrow}^{(S)}k_y\right)\right]\cos\left[\theta_\downarrow^{(S)} + \mathrm{sgn}(E)\left(\vartheta_{k_y\downarrow}^{(S)} + \frac{\lambda^2}{2h^2}k_{F\downarrow}^{(S)}k_y\right)\right]
$$

$$
= \frac{\lambda^2 k_{F\uparrow}^{(S)}k_{F\downarrow}^{(S)}}{h^2}\cos^2\left[\frac{1}{2}\left(\theta_\uparrow^{(S)} - \mathrm{sgn}(E)\,\vartheta_{k_y\uparrow}^{(S)}\right) - \frac{1}{2}\left(\theta_\downarrow^{(S)} + \mathrm{sgn}(E)\,\vartheta_{k_y\downarrow}^{(S)}\right)\right] \tag{35a}
$$

$$
\approx \frac{\lambda^2 k_{F\uparrow}^{(S)}k_{F\downarrow}^{(S)}}{2h^2}\left\{1 + \sin\left[\theta_\uparrow^{(S)} - \mathrm{sgn}(E)\left(\vartheta_{k_y\uparrow}^{(S)} + \frac{\lambda^2}{2h^2}k_{F\uparrow}^{(S)}k_y\right)\right]\right.
$$

$$
\left. \times \sin\left[\theta_\downarrow^{(S)} + \mathrm{sgn}(E)\left(\vartheta_{k_y\downarrow}^{(S)} + \frac{\lambda^2}{2h^2}k_{F\downarrow}^{(S)}k_y\right)\right]\right\}, \tag{35b}
$$

where obtaining the right-hand side of (35b) involves omitting higher-order corrections of the type we have neglected all along. Similarly, the Andreev-approximated form of the characteristic equation (34) for the I-TSF boundary becomes

$$
\cos\left[\theta_\uparrow^{(S)} - \mathrm{sgn}(E)\left(\vartheta_{k_y\uparrow}^{(S)} + \frac{\lambda^2}{2h^2}k_{F\uparrow}^{(S)}k_y\right)\right] = \mathcal{O}\left(\frac{\lambda^2}{h^2}k_{F\uparrow}^{(S)}|k_{F\downarrow}^{(S)}|\frac{m_\downarrow^{(S)}E}{\hbar^2|\mu^{(S)} - \nu_\downarrow^{(S)}|}\right) \approx 0. \tag{36}
$$

We present solutions of Eqs. (35b) and (36), and discuss their physical meaning, in Sec. 3.

Before concluding this subsection, we present an alternative approach to treating the TSF edge that is instructive for our later consideration of the TSF-NSF interface. The system of linear equations arising from requiring the boundary condition (30) for the *Ansatz* (32) can be written as

$$
\begin{pmatrix} \mathcal{D}_\uparrow^{(S)} & \mathcal{C}_{\uparrow\downarrow}^{(S)} \\ \mathcal{C}_{\downarrow\uparrow}^{(S)} & \mathcal{D}_\downarrow^{(S)} \end{pmatrix}\begin{pmatrix} a_{\uparrow++}^{(S)} \\ a_{\uparrow--}^{(S)} \\ a_{\downarrow++}^{(S)} \\ a_{\downarrow-+}^{(S)} \end{pmatrix} = \begin{pmatrix} 0 \\ 0 \\ 0 \\ 0 \end{pmatrix}, \tag{37}
$$

with the $2\times 2$-matrix entries

$$
\mathcal{D}_\uparrow^{(S)} = \begin{pmatrix} u_{\uparrow++}^{(S)} & u_{\uparrow--}^{(S)} \\ v_{\uparrow++}^{(S)} & v_{\uparrow--}^{(S)} \end{pmatrix}, \quad \mathcal{C}_{\downarrow\uparrow}^{(S)} = \begin{pmatrix} \mathcal{J}_{\uparrow++}^{(S)}\begin{pmatrix} u_{\uparrow++}^{(S)} \\ v_{\uparrow++}^{(S)} \end{pmatrix} & \mathcal{J}_{\uparrow--}^{(S)}\begin{pmatrix} u_{\uparrow--}^{(S)} \\ v_{\uparrow--}^{(S)} \end{pmatrix} \end{pmatrix}, \tag{38a}
$$

$$
\mathcal{D}_\downarrow^{(S)} = \begin{pmatrix} u_{\downarrow++}^{(S)} & u_{\downarrow-+}^{(S)} \\ v_{\downarrow++}^{(S)} & v_{\downarrow-+}^{(S)} \end{pmatrix}, \quad \mathcal{C}_{\uparrow\downarrow}^{(S)} = \begin{pmatrix} \mathcal{J}_{\downarrow++}^{(S)}\begin{pmatrix} u_{\downarrow++}^{(S)} \\ v_{\downarrow++}^{(S)} \end{pmatrix} & \mathcal{J}_{\downarrow-+}^{(S)}\begin{pmatrix} u_{\downarrow-+}^{(S)} \\ v_{\downarrow-+}^{(S)} \end{pmatrix} \end{pmatrix}. \tag{38b}
$$

The condition $\det\left(\mathcal{D}_\uparrow^{(S)}\right) = 0$ would yield the familiar [23] chiral-$p$-wave Majorana edge mode for the isolated spin-$\uparrow$ sector, but the coupling between the spin-$\uparrow$ and spin-$\downarrow$ sectors embodied in the matrices $\mathcal{C}_{\uparrow\downarrow}^{(S)}$ and $\mathcal{C}_{\downarrow\uparrow}^{(S)}$ leads to modifications. Straightforward elimination of the coefficients $a_{\downarrow++}^{(S)}$ and $a_{\downarrow-+}^{(S)}$ in Eq. (37) yields the equivalent set

$$
\begin{pmatrix} a_{\downarrow++}^{(S)} \\ a_{\downarrow-+}^{(S)} \end{pmatrix} = -\left[\mathcal{D}_\downarrow^{(S)}\right]^{-1}\mathcal{C}_{\downarrow\uparrow}^{(S)}\begin{pmatrix} a_{\uparrow++}^{(S)} \\ a_{\uparrow--}^{(S)} \end{pmatrix}, \quad \left\{\mathcal{D}_\uparrow^{(S)} - \mathcal{C}_{\uparrow\downarrow}^{(S)}\left[\mathcal{D}_\downarrow^{(S)}\right]^{-1}\mathcal{C}_{\downarrow\uparrow}^{(S)}\right\}\begin{pmatrix} a_{\uparrow++}^{(S)} \\ a_{\uparrow--}^{(S)} \end{pmatrix} = \begin{pmatrix} 0 \\ 0 \end{pmatrix}, \tag{39}
$$

of linear equations, and the characteristic equation for the Andreev-edge-state energy becomes

$$
\det\left(\mathcal{D}_\uparrow^{(S)} - \mathcal{C}_{\uparrow\downarrow}^{(S)}\left[\mathcal{D}_\downarrow^{(S)}\right]^{-1}\mathcal{C}_{\downarrow\uparrow}^{(S)}\right) = 0. \tag{40}
$$

Considering the matrices depending on spin-$\downarrow$ Nambu-spinor amplitudes, we find

$$\mathcal{C}_{\uparrow\downarrow}^{(S)}\left[\mathcal{D}_{\downarrow}^{(S)}\right]^{-1} = \frac{1}{2h}\begin{pmatrix} \lambda\left[\frac{i}{2}\left(k_{\downarrow++}^{(S)} + k_{\downarrow-+}^{(S)}\right) + k_y\right] & -\Delta^{(S)} \\ -\left(\Delta^{(S)}\right)^* & \lambda\left[\frac{i}{2}\left(k_{\downarrow++}^{(S)} + k_{\downarrow-+}^{(S)}\right) - k_y\right] \end{pmatrix}$$

$$+ \mathcal{O}\left(\frac{\lambda k_{F\downarrow}^{(S)}}{h} \frac{m_{\downarrow}^{(S)}\left|\Delta_{\downarrow}^{(S)}\right|}{\hbar^2\left|k_{F\downarrow}^{(S)}\right|^2}\right) \quad (41a)$$

$$\approx \frac{1}{2h}\begin{pmatrix} -\lambda\left(\left|k_{F\downarrow}^{(S)}\right| - k_y\right) & 0 \\ 0 & -\lambda\left(\left|k_{F\downarrow}^{(S)}\right| + k_y\right) \end{pmatrix}, \quad (41b)$$

where the Andreev approximation has been employed in the steps leading to Eq. (41b). Using the form (41b) in (40) and retaining only leading-order corrections in the spin-orbit-coupling strength yields the previously obtained characteristic equation (36). While ultimately giving the same result, the alternative approach required only the manipulation of $2 \times 2$ matrices instead of considering the full $4 \times 4$-matrix determinant arising from the condition (37). Although either way was possible to be pursued successfully in the case of the TSF edge, the analogous dimensional reduction of the $8 \times 8$ problem emerging for the TSF-NSF interface (discussed in the following subsection 2.3) to a more easily tractable $4 \times 4$ system of linear equations will prove crucial for obtaining analytical results in that situation. Furthermore, writing the characteristic equation in the form (40) makes the modifications arising from the coupling between spin-$\uparrow$ and spin-$\downarrow$ sectors more explicit, thus also enabling a more systematic approach to introducing approximations.

## 2.3 SS′ interface — Josephson junction

Describing the SS′ interface located at $x = 0$ requires matching superpositions of evanescent-quasiparticle excitations from the two adjoining regions as per the conditions

$$\begin{pmatrix} u_{\uparrow}^{(S)}(0) \\ v_{\uparrow}^{(S)}(0) \\ u_{\downarrow}^{(S)}(0) \\ v_{\downarrow}^{(S)}(0) \end{pmatrix} = \begin{pmatrix} u_{\uparrow}^{(S')}(0) \\ v_{\uparrow}^{(S')}(0) \\ u_{\downarrow}^{(S')}(0) \\ v_{\downarrow}^{(S')}(0) \end{pmatrix}, \quad \frac{d}{dx}\begin{pmatrix} u_{\uparrow}^{(S)}(x) \\ v_{\uparrow}^{(S)}(x) \\ u_{\downarrow}^{(S)}(x) \\ v_{\downarrow}^{(S)}(x) \end{pmatrix}\Bigg|_{x=0} = \frac{d}{dx}\begin{pmatrix} u_{\uparrow}^{(S')}(x) \\ v_{\uparrow}^{(S')}(x) \\ u_{\downarrow}^{(S')}(x) \\ v_{\downarrow}^{(S')}(x) \end{pmatrix}\Bigg|_{x=0}. \quad (42)$$

Here the general *Ansatz* (7) for the S [S′] part occupying the half-space $x < 0$ [$x > 0$] contains the four Nambu spinors with $\mathbf{k}_{\zeta} = \mathbf{k}_{\sigma\tau\alpha}^{(S)} \equiv \left(k_{\sigma\tau\alpha}^{(S)}, k_y\right)$ for which $\mathfrak{Im}\left(k_{\sigma\tau\alpha}^{(S)}\right) < 0$ $\left[\mathfrak{Im}\left(k_{\sigma\tau\alpha}^{(S')}\right) > 0\right]$. As we consider the situation where S′ is a NSF, the form of the superposition shown on the right-hand-side of Eq. (31) applies; except that S′ should replace S in all superscripts. The explicit form of the *Ansatz* for the TSF in the S region reads

$$\begin{pmatrix} u_{\uparrow}^{(S)}(x) \\ v_{\uparrow}^{(S)}(x) \\ u_{\downarrow}^{(S)}(x) \\ v_{\downarrow}^{(S)}(x) \end{pmatrix} = a_{\uparrow+-}^{(S)} \mathcal{M}_{\uparrow+-}^{(S)} \begin{pmatrix} u_{\uparrow+-}^{(S)} \\ v_{\uparrow+-}^{(S)} \end{pmatrix} e^{i k_{\uparrow+-}^{(S)} x} + a_{\uparrow-+}^{(S)} \mathcal{M}_{\uparrow-+}^{(S)} \begin{pmatrix} u_{\uparrow-+}^{(S)} \\ v_{\uparrow-+}^{(S)} \end{pmatrix} e^{i k_{\uparrow-+}^{(S)} x}$$

$$+ a_{\downarrow+-}^{(S)} \mathcal{M}_{\downarrow+-}^{(S)} \begin{pmatrix} u_{\downarrow+-}^{(S)} \\ v_{\downarrow+-}^{(S)} \end{pmatrix} e^{i k_{\downarrow+-}^{(S)} x} + a_{\downarrow--}^{(S)} \mathcal{M}_{\downarrow--}^{(S)} \begin{pmatrix} u_{\downarrow--}^{(S)} \\ v_{\downarrow--}^{(S)} \end{pmatrix} e^{i k_{\downarrow--}^{(S)} x}. \quad (43)$$

Imposing the matching condition (42) on the *Ansätze* for evanescent-quasiparticle excitations in the TSF and NSF regions yields a homogeneous system of 8 linear equations for the 8 superposition coefficients $a^{(R)}_{\sigma\tau\alpha}$. Similarly to the approach discussed at the end of Sec. 2.2 for treating the TSF edge, the unwieldy system of equations arising from matching at the SS′ interface can be recast in an equivalent $4 \times 4$ form;

$$
\begin{pmatrix} a^{(S)}_{\downarrow+-} \\ a^{(S)}_{\downarrow--} \\ a^{(S')}_{\downarrow++} \\ a^{(S')}_{\downarrow--} \end{pmatrix} = -\left[\mathcal{D}^{(SS')}_{\downarrow}\right]^{-1} \mathcal{C}^{(SS')}_{\downarrow\uparrow} \begin{pmatrix} a^{(S)}_{\uparrow+-} \\ a^{(S)}_{\uparrow-+} \\ a^{(S')}_{\uparrow++} \\ a^{(S')}_{\uparrow--} \end{pmatrix}, \quad \left\{\mathcal{D}^{(SS')}_{\uparrow} - \mathcal{C}^{(SS')}_{\uparrow\downarrow}\left[\mathcal{D}^{(SS')}_{\downarrow}\right]^{-1} \mathcal{C}^{(SS')}_{\downarrow\uparrow}\right\} \begin{pmatrix} a^{(S)}_{\uparrow+-} \\ a^{(S)}_{\uparrow-+} \\ a^{(S')}_{\uparrow++} \\ a^{(S')}_{\uparrow--} \end{pmatrix} = \begin{pmatrix} 0 \\ 0 \\ 0 \\ 0 \end{pmatrix},
$$
(44)

with the matrices

$$
\mathcal{D}^{(SS')}_{\uparrow} = \begin{pmatrix}
u^{(S)}_{\uparrow+-} & u^{(S)}_{\uparrow-+} & -u^{(S')}_{\uparrow++} & -u^{(S')}_{\uparrow--} \\
v^{(S)}_{\uparrow+-} & v^{(S)}_{\uparrow-+} & -v^{(S')}_{\uparrow++} & -v^{(S')}_{\uparrow--} \\
k^{(S)}_{\uparrow+-} u^{(S)}_{\uparrow+-} & k^{(S)}_{\uparrow-+} u^{(S)}_{\uparrow-+} & -k^{(S')}_{\uparrow++} u^{(S')}_{\uparrow++} & -k^{(S')}_{\uparrow--} u^{(S')}_{\uparrow--} \\
k^{(S)}_{\uparrow+-} v^{(S)}_{\uparrow+-} & k^{(S)}_{\uparrow-+} v^{(S)}_{\uparrow-+} & -k^{(S')}_{\uparrow++} v^{(S')}_{\uparrow++} & -k^{(S')}_{\uparrow--} v^{(S')}_{\uparrow--}
\end{pmatrix},
$$
(45a)

$$
\mathcal{D}^{(SS')}_{\downarrow} = \begin{pmatrix}
u^{(S)}_{\downarrow+-} & u^{(S)}_{\downarrow--} & -u^{(S')}_{\downarrow++} & -u^{(S')}_{\downarrow--} \\
v^{(S)}_{\downarrow+-} & v^{(S)}_{\downarrow--} & -v^{(S')}_{\downarrow++} & -v^{(S')}_{\downarrow--} \\
k^{(S)}_{\downarrow+-} u^{(S)}_{\downarrow+-} & k^{(S)}_{\downarrow--} u^{(S)}_{\downarrow--} & -k^{(S')}_{\downarrow++} u^{(S')}_{\downarrow++} & -k^{(S')}_{\downarrow--} u^{(S')}_{\downarrow--} \\
k^{(S)}_{\downarrow+-} v^{(S)}_{\downarrow+-} & k^{(S)}_{\downarrow--} v^{(S)}_{\downarrow--} & -k^{(S')}_{\downarrow++} v^{(S')}_{\downarrow++} & -k^{(S')}_{\downarrow--} v^{(S')}_{\downarrow--}
\end{pmatrix},
$$
(45b)

$$
\mathcal{C}^{(SS')}_{\downarrow\uparrow} = \begin{pmatrix}
\mathcal{J}^{(S)}_{\uparrow+-}\begin{pmatrix} u^{(S)}_{\uparrow+-} \\ v^{(S)}_{\uparrow+-} \end{pmatrix} & \mathcal{J}^{(S)}_{\uparrow-+}\begin{pmatrix} u^{(S)}_{\uparrow-+} \\ v^{(S)}_{\uparrow-+} \end{pmatrix} & -\mathcal{J}^{(S')}_{\uparrow++}\begin{pmatrix} u^{(S')}_{\uparrow++} \\ v^{(S')}_{\uparrow++} \end{pmatrix} & -\mathcal{J}^{(S')}_{\uparrow--}\begin{pmatrix} u^{(S')}_{\uparrow--} \\ v^{(S')}_{\uparrow--} \end{pmatrix} \\
k^{(S)}_{\uparrow+-}\mathcal{J}^{(S)}_{\uparrow+-}\begin{pmatrix} u^{(S)}_{\uparrow+-} \\ v^{(S)}_{\uparrow+-} \end{pmatrix} & k^{(S)}_{\uparrow-+}\mathcal{J}^{(S)}_{\uparrow-+}\begin{pmatrix} u^{(S)}_{\uparrow-+} \\ v^{(S)}_{\uparrow-+} \end{pmatrix} & -k^{(S')}_{\uparrow++}\mathcal{J}^{(S')}_{\uparrow++}\begin{pmatrix} u^{(S')}_{\uparrow++} \\ v^{(S')}_{\uparrow++} \end{pmatrix} & -k^{(S')}_{\uparrow--}\mathcal{J}^{(S')}_{\uparrow--}\begin{pmatrix} u^{(S')}_{\uparrow--} \\ v^{(S')}_{\uparrow--} \end{pmatrix}
\end{pmatrix},
$$
(45c)

$$
\mathcal{C}^{(SS')}_{\uparrow\downarrow} = \begin{pmatrix}
\mathcal{J}^{(S)}_{\downarrow+-}\begin{pmatrix} u^{(S)}_{\downarrow+-} \\ v^{(S)}_{\downarrow+-} \end{pmatrix} & \mathcal{J}^{(S)}_{\downarrow--}\begin{pmatrix} u^{(S)}_{\downarrow--} \\ v^{(S)}_{\downarrow--} \end{pmatrix} & -\mathcal{J}^{(S')}_{\downarrow++}\begin{pmatrix} u^{(S')}_{\downarrow++} \\ v^{(S')}_{\downarrow++} \end{pmatrix} & -\mathcal{J}^{(S')}_{\downarrow--}\begin{pmatrix} u^{(S')}_{\downarrow--} \\ v^{(S')}_{\downarrow--} \end{pmatrix} \\
k^{(S)}_{\downarrow+-}\mathcal{J}^{(S)}_{\downarrow+-}\begin{pmatrix} u^{(S)}_{\downarrow+-} \\ v^{(S)}_{\downarrow+-} \end{pmatrix} & k^{(S)}_{\downarrow--}\mathcal{J}^{(S)}_{\downarrow--}\begin{pmatrix} u^{(S)}_{\downarrow--} \\ v^{(S)}_{\downarrow--} \end{pmatrix} & -k^{(S')}_{\downarrow++}\mathcal{J}^{(S')}_{\downarrow++}\begin{pmatrix} u^{(S')}_{\downarrow++} \\ v^{(S')}_{\downarrow++} \end{pmatrix} & -k^{(S')}_{\downarrow--}\mathcal{J}^{(S')}_{\downarrow--}\begin{pmatrix} u^{(S')}_{\downarrow--} \\ v^{(S')}_{\downarrow--} \end{pmatrix}
\end{pmatrix}.
$$
(45d)

Alternatively to (44), the original $8 \times 8$ system of linear equations can be formally expressed in terms of an equivalent other set of $4 \times 4$ relations

$$
\begin{pmatrix} a^{(S)}_{\uparrow+-} \\ a^{(S)}_{\uparrow-+} \\ a^{(S')}_{\uparrow++} \\ a^{(S')}_{\uparrow--} \end{pmatrix} = -\left[\mathcal{D}^{(SS')}_{\uparrow}\right]^{-1} \mathcal{C}^{(SS')}_{\uparrow\downarrow} \begin{pmatrix} a^{(S)}_{\downarrow+-} \\ a^{(S)}_{\downarrow--} \\ a^{(S')}_{\downarrow++} \\ a^{(S')}_{\downarrow--} \end{pmatrix}, \quad \left\{\mathcal{D}^{(SS')}_{\downarrow} - \mathcal{C}^{(SS')}_{\downarrow\uparrow}\left[\mathcal{D}^{(SS')}_{\uparrow}\right]^{-1} \mathcal{C}^{(SS')}_{\uparrow\downarrow}\right\} \begin{pmatrix} a^{(S)}_{\downarrow+-} \\ a^{(S)}_{\downarrow--} \\ a^{(S')}_{\downarrow++} \\ a^{(S')}_{\downarrow--} \end{pmatrix} = \begin{pmatrix} 0 \\ 0 \\ 0 \\ 0 \end{pmatrix}.
$$
(46)

The form of the matrices $\mathcal{D}_\sigma^{(SS')}$ could be interpreted as arising from matching conditions for large Nambu-spinor components of the isolated spin-$\sigma$ subsystems, with each of these constituting a junction between chiral-$p$-wave superfluids with the same chirality on both sides of the interface but different pair-potential magnitudes. This analogy is most immediate for the spin-$\uparrow$ sector whose effective chemical potential $\mu - v_\uparrow^{(R)}$ is positive in both the S and S$'$ regions, and for which the interface therefore constitutes a genuine Josephson junction. In contrast, the fact that $\mu - v_\downarrow^{(S)} < 0 < \mu - v_\downarrow^{(S')}$ suggests that the TSF-NSF interface acts like a wall for spin-$\downarrow$ quasiparticles from the NSF region, raising the possibility for a Majorana edge state to emerge.

The following subsection 2.3.1 formalises the description of the SS$'$ interface in terms of chiral-$p$-wave junctions of fully separated spin subsystems. To provide a reference point for comparison, as well as further relevant background information, we derive the Andreev bound states at the most general realization of an interface between two arbitrary spinless chiral-$p$-wave systems in Appendix A. Results presented there generalize those given in related previous works [24, 43–46] to the situation where the order-parameter magnitudes are different in the two regions.

While it may be tempting to limit considerations to the matrices $\mathcal{D}_\sigma^{(SS')}$ representing individual spin-$\sigma$ subsystems, such an approach neglects the coupling between spin-$\uparrow$ and spin-$\downarrow$ sectors embodied in the matrices $\mathcal{C}_{\sigma\bar{\sigma}}^{(SS')}$. In effect, our SS$'$ hybrid system should be most appropriately thought of as a Josephson junction between two two-band superconductors where the two bands are distinguished by the spin projection $\sigma \in \{\uparrow, \downarrow\}$. Existing theoretical descriptions of two-band-superconductor junctions rely heavily on numerics [61]. In sections 2.3.2 and 2.3.3, we develop approximate analytical methods to explore whether and how the residual coupling between opposite-spin sectors modifies the predictions obtained within the simple picture from Sec. 2.3.1. Physical consequences for the Andreev-bound-state spectrum at the TSF-NSF interface are discussed in detail in Sec. 4.

### 2.3.1 Description in terms of completely decoupled spin subsystems

On the most elementary level, the energies of interface-localized Andreev bound states within individual spin-$\sigma$ sectors can be found as solutions of the characteristic equations

$$\det\left(\mathcal{D}_\uparrow^{(SS')}\right) \approx 4\left(k_{F\uparrow}^{(SS')}\right)^2 \left(u_{\uparrow+-}^{(S)} v_{\uparrow--}^{(S')} - u_{\uparrow--}^{(S')} v_{\uparrow+-}^{(S)}\right)\left(u_{\uparrow-+}^{(S)} v_{\uparrow++}^{(S')} - u_{\uparrow++}^{(S')} v_{\uparrow-+}^{(S)}\right) = 0, \tag{47a}$$

$$\det\left(\mathcal{D}_\downarrow^{(SS')}\right) \approx \left[\left|k_{F\downarrow}^{(S)}\right|^2 + \left(k_{F\downarrow}^{(S')}\right)^2\right]\left(u_{\downarrow+-}^{(S)} v_{\downarrow--}^{(S)} - u_{\downarrow--}^{(S)} v_{\downarrow+-}^{(S)}\right)\left(u_{\downarrow--}^{(S')} v_{\downarrow++}^{(S')} - u_{\downarrow++}^{(S')} v_{\downarrow--}^{(S')}\right) = 0. \tag{47b}$$

Here the determinants of matrices $\mathcal{D}_\uparrow^{(SS')}$ and $\mathcal{D}_\downarrow^{(SS')}$ given explicitly in Eqs. (45a) and (45b), respectively, have been calculated to leading order in the Andreev approximation (27) where

$$k_{\uparrow\tau\alpha}^{(R)} \approx \alpha\, k_{F\uparrow}^{(SS')}, \quad \text{with} \quad k_{F\uparrow}^{(SS')} \equiv \sqrt{2m(\mu + h)/\hbar^2}, \tag{48a}$$

$$k_{\downarrow\tau\alpha}^{(S')} \approx \alpha\, k_{F\downarrow}^{(S')}, \quad \text{and} \quad k_{\downarrow\tau\alpha}^{(S)} \approx \alpha\, i\left|k_{F\downarrow}^{(S)}\right|. \tag{48b}$$

Using also the Andreev-approximation expressions (29) for Nambu-spinor components in the separated-spin-$\uparrow$-sector characteristic equation (47a) yields the $Z = 0$, $\gamma^{(S)} = \gamma^{(S')}$ limit of the general spinless-chiral-$p$-wave-junction characteristic equation derived in Appendix A [see Eq. (A.10)], which reads explicitly

$$\cos\left(\theta_\uparrow^{(S')} + \theta_\uparrow^{(S)}\right) = \cos\left(\varphi^{(S')} - \varphi^{(S)}\right). \tag{49}$$

Thus, within the Andreev-approximation treatment, the completely decoupled spin-$\uparrow$ sector of the considered realization of a TSF-NSF interface constitutes a fully transparent Josephson

junction between equal-chirality *p*-wave superfluids with different pair-potential magnitude. Appendix A provides a more detailed discussion of such a system's physical properties, including a comparison with those of a conventional s-wave junction.

The fact that $\mu - v_\downarrow^{(S)} < 0$ implies that the separated-spin-↓-sector characteristic equation (47b) can only be satisfied when the relation

$$u_{\downarrow--}^{(S')} v_{\downarrow++}^{(S')} - u_{\downarrow++}^{(S')} v_{\downarrow--}^{(S')} = 0 \tag{50}$$

holds, which is the defining equation for the surface bound state of an unconventional superconductor [48,62], including the Majorana edge mode in a chiral-*p*-wave superfluid [23]. In particular, inserting the Andreev-approximation expressions (29) for Nambu-spinor amplitudes appearing in Eq. (50) transforms the latter into

$$\cos\left[ \theta_\downarrow^{(S')} + \mathrm{sgn}(E)\, \vartheta_{k_y\downarrow}^{(S')} \right] = 0. \tag{51}$$

We discuss in more detail in Appendix A how the Majorana-mode energy dispersion emerges as the solution of characteristic equations that are of the form (51), and the Majorana-fermion character of the corresponding eigenstates is demonstrated in Appendix B. Thus the intuitive expectation that a TSF-NSF interface constitutes a boundary for the spin-↓-sector quasiparticles in the NSF region is borne out within the separated-spin-sector description.

### 2.3.2 Opposite-spin-coupling modifications for the spin-↓ subsystem

We base our investigation of modifications generated in the spin-↓-sector characteristic equation by the coupling to the spin-↑ degrees of freedom on the relations (46). Requiring that the system of linear equations on the right-hand side of Eq. (46) has a nontrivial solution leads to the characteristic equation

$$\det\left( \mathcal{D}_\downarrow^{(SS')} - \mathcal{C}_{\downarrow\uparrow}^{(SS')} \left[ \mathcal{D}_\uparrow^{(SS')} \right]^{-1} \mathcal{C}_{\uparrow\downarrow}^{(SS')} \right) = 0. \tag{52}$$

As the spin-↑ sector is deeply in the BCS regime and $h \sim \mu$ as per Eq. (3), it is possible to neglect $|\Delta^{(R)}|/h$ corrections in the matrices $\mathcal{J}_{\uparrow\tau\alpha}^{(R)}$. Using also the approximation (48a), we find

$$\mathcal{C}_{\downarrow\uparrow}^{(SS')} \left[ \mathcal{D}_\uparrow^{(SS')} \right]^{-1} \approx \frac{\lambda}{2h} \begin{pmatrix} -k_y & 0 & i & 0 \\ 0 & k_y & 0 & i \\ i\left(k_{F\uparrow}^{(SS')}\right)^2 & 0 & -k_y & 0 \\ 0 & i\left(k_{F\uparrow}^{(SS')}\right)^2 & 0 & k_y \end{pmatrix}. \tag{53}$$

Utilizing the expression (53) in the term providing a correction to $\mathcal{D}_\downarrow^{(SS')}$ in (52) yields

$$\mathcal{C}_{\downarrow\uparrow}^{(SS')} \left[ \mathcal{D}_\uparrow^{(SS')} \right]^{-1} \mathcal{C}_{\uparrow\downarrow}^{(SS')} = -\sqrt{\frac{m\lambda^2}{\hbar^2 h}}\, \mathcal{A}_\downarrow^{(SS')} - \frac{m\lambda^2}{\hbar^2 h}\, \mathcal{B}_\downarrow^{(SS')}. \tag{54}$$

The relation $h \sim \mu$ noted already above also implies that the spin-↓ wave numbers $k_{\downarrow\tau\alpha}^{(R)}$ appearing in the $4 \times 4$ matrices $\mathcal{D}_\downarrow^{(SS')}$, $\mathcal{A}_\downarrow^{(SS')}$ and $\mathcal{B}_\downarrow^{(SS')}$ are small quantities. Specifically, we have $k_{\downarrow\tau\alpha}^{(R)} = \mathcal{O}\left( \sqrt{m_\downarrow^{(R)} |\Delta^{(R)}|^2 / (\hbar^2 h)} \right)$ and, consequently, $k_{\downarrow\tau\alpha}^{(R)} / k_{F\uparrow}^{(SS')} = \mathcal{O}(|\Delta^{(R)}|/h)$. Neglecting terms $\mathcal{O}(|\Delta^{(R)}|/h)$ as well as $\mathcal{O}\left( [k_y / k_{\downarrow\tau\alpha}^{(R)}]^2 \right)$, consistent with the Andreev approximation as

applied to the spin-$\uparrow$ sector [see the discussion preceding Eq. (53)], then yields

$$
\mathcal{A}_{\downarrow}^{(SS')} \approx i\,\sqrt{\frac{m}{\hbar^2 h}}\begin{pmatrix} 0 & 0 & 0 & 0 \\ 0 & 0 & 0 & 0 \\ \Delta^{(S)}v_{\downarrow+-}^{(S)} & \Delta^{(S)}v_{\downarrow--}^{(S)} & -\Delta^{(S')}v_{\downarrow++}^{(S')} & -\Delta^{(S')}v_{\downarrow--}^{(S')} \\ \left(\Delta^{(S)}\right)^*u_{\downarrow+-}^{(S)} & \left(\Delta^{(S)}\right)^*u_{\downarrow--}^{(S)} & -\left(\Delta^{(S')}\right)^*u_{\downarrow++}^{(S')} & -\left(\Delta^{(S')}\right)^*u_{\downarrow--}^{(S')} \end{pmatrix}, \quad (55a)
$$

$$
\mathcal{B}_{\downarrow}^{(SS')} \approx \begin{pmatrix} 0 & 0 & 0 & 0 \\ 0 & 0 & 0 & 0 \\ \begin{aligned}u_{\downarrow+-}^{(S)}&\left\{k_{\downarrow+-}^{(S)}-ik_y\right.\\&\left.\times\left[1-\left(\frac{k_{\downarrow+-}^{(S)}}{k_{F\uparrow}^{(SS')}}\right)^2\right]\right\}\end{aligned} & \begin{aligned}u_{\downarrow--}^{(S)}&\left\{k_{\downarrow--}^{(S)}-ik_y\right.\\&\left.\times\left[1-\left(\frac{k_{\downarrow--}^{(S)}}{k_{F\uparrow}^{(SS')}}\right)^2\right]\right\}\end{aligned} & \begin{aligned}-u_{\downarrow++}^{(S')}&\left\{k_{\downarrow++}^{(S')}-ik_y\right.\\&\left.\times\left[1-\left(\frac{k_{\downarrow++}^{(S')}}{k_{F\uparrow}^{(SS')}}\right)^2\right]\right\}\end{aligned} & \begin{aligned}-u_{\downarrow--}^{(S')}&\left\{k_{\downarrow--}^{(S')}-ik_y\right.\\&\left.\times\left[1-\left(\frac{k_{\downarrow--}^{(S')}}{k_{F\uparrow}^{(SS')}}\right)^2\right]\right\}\end{aligned} \\ \begin{aligned}v_{\downarrow+-}^{(S)}&\left\{k_{\downarrow+-}^{(S)}+ik_y\right.\\&\left.\times\left[1-\left(\frac{k_{\downarrow+-}^{(S)}}{k_{F\uparrow}^{(SS')}}\right)^2\right]\right\}\end{aligned} & \begin{aligned}v_{\downarrow--}^{(S)}&\left\{k_{\downarrow--}^{(S)}+ik_y\right.\\&\left.\times\left[1-\left(\frac{k_{\downarrow--}^{(S)}}{k_{F\uparrow}^{(SS')}}\right)^2\right]\right\}\end{aligned} & \begin{aligned}-v_{\downarrow++}^{(S')}&\left\{k_{\downarrow++}^{(S')}+ik_y\right.\\&\left.\times\left[1-\left(\frac{k_{\downarrow++}^{(S')}}{k_{F\uparrow}^{(SS')}}\right)^2\right]\right\}\end{aligned} & \begin{aligned}-v_{\downarrow--}^{(S')}&\left\{k_{\downarrow--}^{(S')}+ik_y\right.\\&\left.\times\left[1-\left(\frac{k_{\downarrow--}^{(S')}}{k_{F\uparrow}^{(SS')}}\right)^2\right]\right\}\end{aligned} \end{pmatrix}. \quad (55b)
$$

Subleading-order corrections to factors multiplying $k_y$ have been retained in $\mathcal{B}_{\downarrow}^{(SS')}$ for illustration, as these contribute a leading-order correction in the limit of small spin-orbit coupling. The nonzero entries of $\mathcal{A}_{\downarrow}^{(SS')}$ as given in Eq. (55a) have a magnitude $\mathcal{O}\left(\sqrt{m\left|\Delta^{(R)}\right|^2/(\hbar^2 h)}\right)$, which is the same as that of the $k_{\downarrow\tau\alpha}^{(R)}$ prefactors of entries in the bottom two rows of $\mathcal{D}_{\downarrow}^{(SS')}$ [see Eq. (45b)]. However, the additional factor $\sqrt{m\lambda^2/(\hbar^2 h)}\ll 1$ in front of $\mathcal{A}_{\downarrow}^{(SS')}$ in Eq. (54) renders this term's contribution negligible within the Andreev approximation [see Eq. (28) and discussion below]. In contrast, the correction term involving the matrix $\mathcal{B}_{\downarrow}^{(SS')}$ in Eq. (54) formally constitutes a leading-order correction in small spin-orbit-coupling magnitude to $\mathcal{D}_{\downarrow}^{(SS')}$. Retaining this term when evaluating (52) would yield the characteristic equation

$$
\cos\left[\theta_{\downarrow}^{(S')}+\mathrm{sgn}(E)\left(\vartheta_{k_y\downarrow}^{(S')}+\frac{\lambda^2}{2h^2}k_{F\downarrow}^{(S')}k_y\right)\right]=0\,, \quad (56)
$$

which is of the same general form as Eq. (36) found for the I-TSF edge state. However, because of the parametric smallness of $k_{F\downarrow}^{(S')}=\mathcal{O}\left(\sqrt{m_{\downarrow}^{(S')}\left|\Delta^{(S')}\right|^2/(\hbar^2 h)}\right)$ within our particular realization of the TSF-NSF interface, overall consistency requires that the $\lambda^2$-dependent correction in Eq. (56) is also neglected. As a consequence, to leading order in the Andreev approximation, the coupling to the spin-$\uparrow$ subsystem does not affect the properties of the spin-$\downarrow$ Andreev bound state formed at the SS' interface, and the characteristic equation yielding its energy dispersion is given by the result Eq. (51) obtained in the limit where the spin subsectors are considered to be uncoupled.

### 2.3.3 Opposite-spin-coupling modifications for the spin-$\uparrow$ subsystem

We now turn to discussing properties of the Josephson junction realized in the spin-$\uparrow$ sector, utilizing the relations shown in Eqs. (44). The characteristic equation for interface-localized bound states is

$$
\det\left(\mathcal{D}_{\uparrow}^{(SS')}-\mathcal{C}_{\uparrow\downarrow}^{(SS')}\left[\mathcal{D}_{\downarrow}^{(SS')}\right]^{-1}\mathcal{C}_{\downarrow\uparrow}^{(SS')}\right)=0\,. \quad (57)
$$

As we are interested in obtaining corrections to the Andreev-approximated form of the characteristic equation (47a), we can neglect $|\Delta^{(R)}|/h$ corrections in the matrices $\mathcal{J}^{(R)}_{\downarrow\tau\alpha}$ entering $\mathcal{C}^{(SS')}_{\uparrow\downarrow}$. A straightforward calculation using the relations from Eq. (48b) then yields

$$
\mathcal{C}^{(SS')}_{\uparrow\downarrow}\left[\mathcal{D}^{(SS')}_{\downarrow}\right]^{-1} \approx \frac{\lambda}{2h}\begin{pmatrix} k_y & 0 & i & 0 \\ 0 & -k_y & 0 & i \\ 0 & 0 & \left|k^{(S)}_{F\downarrow}\right|+k_y & 0 \\ 0 & 0 & 0 & \left|k^{(S)}_{F\downarrow}\right|-k_y \end{pmatrix} + \frac{\lambda\, k^{(S')}_{F\downarrow}}{2h}\begin{pmatrix} 0 & 0 & 0 & 0 \\ 0 & 0 & 0 & 0 \\ \left|k^{(S)}_{F\downarrow}\right|a & \left|k^{(S)}_{F\downarrow}\right|b & -i\,a & -i\,b \\ \left|k^{(S)}_{F\downarrow}\right|c & -\left|k^{(S)}_{F\downarrow}\right|a & -i\,c & i\,a \end{pmatrix},
$$
(58)

with the abbreviations

$$
a = \frac{u^{(S')}_{\downarrow--}\,v^{(S')}_{\downarrow++} + u^{(S')}_{\downarrow++}\,v^{(S')}_{\downarrow--}}{u^{(S')}_{\downarrow--}\,v^{(S')}_{\downarrow++} - u^{(S')}_{\downarrow++}\,v^{(S')}_{\downarrow--}}, \quad b = -\frac{2\,u^{(S')}_{\downarrow--}\,u^{(S')}_{\downarrow++}}{u^{(S')}_{\downarrow--}\,v^{(S')}_{\downarrow++} - u^{(S')}_{\downarrow++}\,v^{(S')}_{\downarrow--}}, \quad c = \frac{2\,v^{(S')}_{\downarrow++}\,v^{(S')}_{\downarrow--}}{u^{(S')}_{\downarrow--}\,v^{(S')}_{\downarrow++} - u^{(S')}_{\downarrow++}\,v^{(S')}_{\downarrow--}}.
$$
(59)

The factors $a$, $b$ and $c$ entering Eq. (58) depend on the unknown energy of the spin-$\uparrow$-sector Andreev bound state via the $E$-dependence of Nambu-spinor components [see general expressions from Eqs. (23) or their Andreev-approximated forms (29)]. Also, the denominator appearing in each of these quantities [see Eq. (59)] coincides with the expression on the left-hand side of Eq. (50) that is the characteristic equation for the energy of the spin-$\downarrow$-sector Majorana edge state. As a result, a divergence occurs when $E$ coincides with the spin-$\downarrow$ Majorana-edge-state dispersion. We therefore limit further discussion to the range of $k_y$ values within which the energy of the spin-$\uparrow$-sector Andreev bound state is well-separated from that of the emergent Majorana mode in the spin-$\downarrow$ sector, as there the quantities $a$, $b$ and $c$ remain well-defined and all have a magnitude of $\mathcal{O}(1)$. Due to the parametric smallness of spin-$\downarrow$ wave vectors, the contribution of the second term in Eq. (58) then becomes negligible.

Using the first term on the right-hand side of Eq. (58) for $\mathcal{C}^{(SS')}_{\uparrow\downarrow}\left[\mathcal{D}^{(SS')}_{\downarrow}\right]^{-1}$ in the characteristic equation (57) turns out to only yield corrections that are small within the Andreev approximation for spin-$\uparrow$-sector quantities. Hence, at least to that level of approximation and for the assumed range of $k_y$ where $E$ is far from the Majorana-edge-state energy of the spin-$\downarrow$ sector, the description of the spin-$\uparrow$ system within the completely decoupled limit remains unmodified.

## 3 Results & discussion I: Edge states at I-NSF and I-TSF interfaces

We have derived characteristic equations for the Andreev-bound-state energies at the edge of a spin-orbit-coupled polarized 2D Fermi superfluid for when it is in the NSF phase [Eq. (35b)] or the TSF phase [Eq. (36)]. These results are accurate to leading order in the small parameter $m\lambda^2/(\hbar^2 h)$, as our formalism fundamentally relies on the assumption of small-enough spin-orbit-coupling magnitude. We also employed the familiar Andreev approximation [59], which for our system of interest implies the relation (28). We now present solutions of these characteristic equations and discuss physical properties of the associated Andreev bound states.

The characteristic equation (35b) for edge states of the NSF is formally analogous to the general expression

$$
\cos\left[\theta^{(S')} + \text{sgn}(E)\,\gamma^{(S')}\,\vartheta^{(S')}_{k_y}\right]\cos\left[\theta^{(S)} - \text{sgn}(E)\,\gamma^{(S)}\,\vartheta^{(S)}_{k_y}\right]
$$
(60)

$$
= \frac{T}{2-T}\left\{\cos\left(\varphi^{(S')} - \varphi^{(S)}\right) + \sin\left[\theta^{(S')} + \text{sgn}(E)\gamma^{(S')}\vartheta^{(S')}_{k_y}\right]\sin\left[\theta^{(S)} - \text{sgn}(E)\gamma^{(S)}\vartheta^{(S)}_{k_y}\right]\right\},
$$

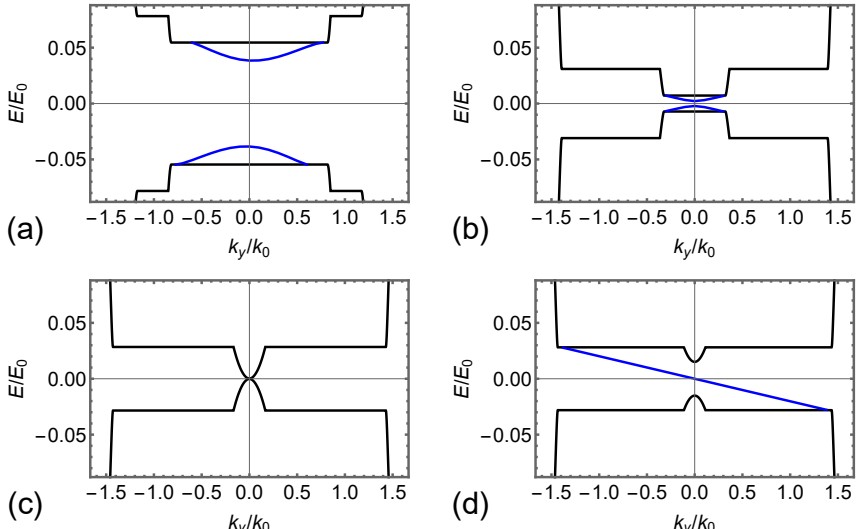

Figure 3: Energy dispersion of Andreev bound states at an IS interface oriented parallel to the $y$ direction. Panels (a) and (b) depict situations where S is a nontopological superfluid (NSF). Panel (c) corresponds to the topological-transition point where $h = h_c \equiv \sqrt{\mu^2 + \left|\Delta^{(S)}\right|^2}$, and panel (d) is for S being a topological superfluid (TSF). Blue curves show the energy dispersion of Andreev edge states as a function of 2D-wave-vector component $k_y$ parallel to the interface, arising from solution of the characteristic equations (35b) [panels (a) and (b)] and (36) [panel (d)]. Black curves indicate the minimum-energy bound of the quasiparticle-excitation continuum calculated from the $2 \times 2$-projected-theory dispersions (18). The arbitrary units $k_0$ and $E_0$ for wave number and energy are related via $E_0 = \hbar^2 k_0^2/(2m)$. Results shown are for $\lambda k_0/E_0 = 0.2$, $\left|\Delta^{(S)}\right|/E_0 = 0.1$, $\mu/E_0 = 1$ in all panels and $h/E_0 = 0.3$ in (a), 0.9 in (b), $\sqrt{1.01}$ in (c) and 1.02 in (d).

for the equation determining Andreev-bound-state energies at a Josephson (SS′) junction between two spinless chiral $p$-wave superfluids [24, 43–46]. In Eq. (60), the superscript $R \in \{S, S'\}$ labels quantities from the left (S) and right (S′) sides of the junction, respectively, $\gamma^{(R)} = \pm 1$ are the chiralities of the $p$-wave pair potentials, $\varphi^{(R)}$ is the pair-potential phase in region $R$, and $0 \leq T \leq 1$ is the junction's transparency for particle transmission. See Appendix A for a detailed derivation. Setting $\varphi^{(S')} - \varphi^{(S)} \to 0$ and $T/(2 - T) \to \lambda^2 k_{\mathrm{F}\uparrow}^{(S)} k_{\mathrm{F}\downarrow}^{(S)}/(2h^2)$ in (60), together with an adjusted definition of $\vartheta_{k_y}$, yields (35b). Physically, this makes sense, as the opposite-spin subsystems of the NSF, each constituting an effective realization of a spinless chiral-$p$-wave superfluid, are coupled in a way that is similar to a Josephson junction via the finite spin-orbit coupling $\lambda \neq 0$. As it is well-known [44, 45] that the energy of the Andreev bound state at chiral-$p$-wave SS′ junctions with $\varphi^{(S')} - \varphi^{(S)} = 0$ is finite for $k_y \ll k_{\mathrm{F}}$ [see also Fig. 5 in Appendix A], we expect the same for the NSF edge states. Plotting solutions of (35b) confirms our expectation; see Figs. 3(a) and 3(b). As the minimum of the subgap-excitation energy occurs close to (albeit not exactly at) $k_y = 0$, it is useful to note the analytical result

$$|E(k_y = 0)| \approx \frac{\lambda^2 k_{\mathrm{F}\uparrow}^{(S)} k_{\mathrm{F}\downarrow}^{(S)}}{h^2} \left|\Delta^{(S)}\right| \equiv \frac{\lambda k_{\mathrm{F}\uparrow}^{(S)}}{h} \Delta_\downarrow^{(S)}. \tag{61}$$

Thus while the Andreev-edge-state energy for the NSF is finite, it is small because we are focusing on the situation where $\lambda k_{\mathrm{F}\uparrow}^{(S)} \ll h$. Furthermore, it vanishes together with $\Delta_\downarrow^{(S)}$ at the topological transition; see Fig. 3(c).

Turning to the solution of the characteristic equation (36) for the Andreev-edge-state energy of the I-TSF boundary, we find

$$E(k_y) = -\left[1 + \frac{\lambda^2 \left(k_{F\uparrow}^{(S)}\right)^2}{2h^2}\right] \frac{\Delta_\uparrow^{(S)}}{k_{F\uparrow}^{(S)}} k_y \equiv -\left[1 + \frac{\lambda^2 \left(k_{F\uparrow}^{(S)}\right)^2}{2h^2}\right] \frac{\lambda \left|\Delta^{(S)}\right|}{h} k_y \,. \tag{62}$$

Apart from the $\lambda^2$-dependent correction in brackets, the result (62) coincides with the dispersion of the familiar [23–29] Majorana edge mode of a chiral-$p$-wave superconductor, with pertinent parameters from the spin-↑ sector of the TSF. (See the discussion in the last paragraph of Appendix A, as well as the entire Appendix B, for relevant background information.) A specific example is shown in Fig. 3(d). As terms of order $\mathcal{O}(k_y^2/k_{F\uparrow}^2)$ are neglected as part of the Andreev approximation, we do not resolve nonlinear corrections to the topological-edge-state dispersion that should become important when it approaches the quasiparticle-excitation continuum. The TSF-edge mode with dispersion (62) corresponds to the IS-interface Andreev bound state indicated by the solid green line in Fig. 1(a): it is a quasiparticle with dominant-spin-↑ character and propagates in negative-$y$ direction along the interface.

Our analytical results reproduce salient features seen in numerically obtained excitation spectra for our system of interest, e.g., those shown in insets of Fig. 7 from Ref. [33]. While nontopological edge states in superconducting spin-orbit-coupled nanowires have been studied extensively (see Ref. [63] as one of the seminal works and Refs. [64,65] for comprehensive overviews), the existence of the nontopological edge state at the I-NSF boundary in a 2D system has not been discussed so far. Unlike the topologically protected (Majorana) edge state of the I-TSF system, the subgap excitation at the NSF edge is not robust against perturbations. Nevertheless, as it influences the low-energy physics of the I-NSF system, spectroscopic techniques [9,66] should be able to detect it and distinguish it from the Majorana mode.

## 4 Results & discussion II: Bound states at the TSF-NSF interface

Technically, the TSF-NSF interface exhibits all the hallmarks of a complicated multi-band Josephson junction. However, considerable simplifications would arise under the assumption of perfect decoupling for the majority-spin (spin-↑) and minority-spin (spin-↓) sectors within the TSF and NSF phases. In this limit, the interface becomes equivalent to a configuration of two parallel Josephson junctions: one between the chiral-$p$-wave superfluids realized in the spin-↑ sectors of the TSF and NSF regions, and the other one between those of the spin-↓ sectors. The chiral-$p$-wave Josephson junction formed by the spin-↑ degrees of freedom from the TSF and NSF regions is different from previously considered types [24, 43–46] because the order-parameter magnitudes on opposite sides of the interface are different. Furthermore, with the spin-↓ degrees of freedom being Zeeman-energy-quenched in the TSF, the TSF-NSF interface can be expected to act as a wall for the spin-↓ chiral-$p$-wave superfluid in the NSF.

While certainly being attractive for its simplicity, the perfect-decoupling limit potentially fails to describe reality. As described in detail in Sec. 2, the opposite-spin sectors of a spin-orbit-coupled polarized 2D Fermi superfluid are not completely independent, and this residual coupling can have important physical consequences. For example, application of the perfect-decoupling limit to the I-NSF interface would result in the expectation that it hosts a helical edge mode, consisting of the two counter-propagating chiral edge modes arising from individual chiral-$p$-wave superfluids realized with the spin-↑ and spin-↓ sectors. That this is not the case is known from numerical studies and general topological considerations [16, 67]. Our detailed calculations described in Sec. 2.2 and results discussed in Sec. 3 elucidate how the properties of the I-NSF interface are manifestly shaped by the coupling between opposite-spin

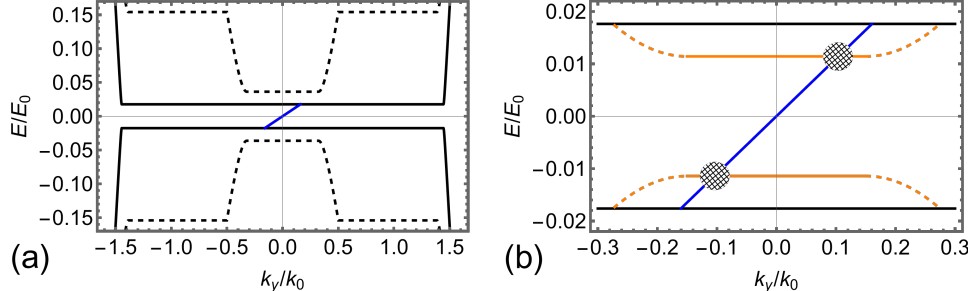

Figure 4: Andreev bound states and quasiparticle-continuum edges at the TSF (S) –
NSF (S′) interface. The interface-localised Andreev bound state with Majorana char-
acter is realised in the minority-spin (spin-↓) sector. Panel (a) shows its energy disper-
sion from Eq. (63) as the diagonal blue line. The solid (dashed) black curves indicate
the boundary of the quasiparticle-excitation continua in the TSF (NSF) region calcu-
lated from the $2 \times 2$-projected-theory dispersions (18). Panel (b) is a zoom-in to the
gap region, including additionally the Andreev bound states from the majority-spin
(spin-↑) sector. Solid orange curves plot the $k_y$-independent dispersions of Eq. (64)
with $\varphi^{(S')} - \varphi^{(S)} = 3\pi/4$. Dashed continuations indicate the expected shape of these
dispersions at larger $k_y$, which lie beyond the approximations of our theory. The
hatched circles cover regions where the Andreev-bound-state energies of both spin
sectors are not well-separated and results obtained within the uncoupled-spin-sector
approach no longer apply. The parameters chosen correspond to the SS'-interface
of Fig. 1(c) where the bulk quasiparticle excitation gap is smaller in the TSF com-
pared to the NSF. The parameter values are $\lambda k_0/E_0 = 0.2$, $\mu/E_0 = 1.0$, $h/E_0 = 1.05$,
$\left|\Delta^{(S)}\right|/E_0 = 0.06$ and $\left|\Delta^{(S')}\right|/E_0 = 0.58$. The arbitrary wave-number and energy
scales $k_0$ and $E_0$ are related via $E_0 = \hbar^2 k_0^2/(2m)$.

sectors of the NSF. It is therefore necessary to investigate whether and how the conclusions de-
rived for the TSF-NSF interface within the naïve picture that assumes perfectly decoupled spin
sectors are altered. This has been the purpose of our careful analysis performed in Sec. 2.3,
and we now discuss our obtained results.

Modifications to the NSF's spin-↓ edge mode at the TSF-NSF interface arising from its cou-
pling to the spin-↑ sector are captured by the characteristic equation (56). This expression
closely resembles the characteristic equation (36) for the TSF edge. However, the parametric
smallness of $k_{\mathrm{F\downarrow}}^{(S')}$ that is inherent in our envisioned realization of the TSF-NSF hybrid system
via a spatially varying $s$-wave pair potential [see Eqs. (2) and (3), as well as the associated dis-
cussion in Sec. 1] implies that the $\lambda^2$-dependent correction term in (56) should be neglected
within the Andreev approximation, yielding the characteristic equation (51) obtained for the
completely decoupled spin-↓ sector. Solving (51) to find the energy dispersion for the edge
state localised at the SS′ junction yields

$$E_\downarrow(k_y) = \frac{\Delta_\downarrow^{(S')}}{k_{\mathrm{F\downarrow}}^{(S')}} k_y \equiv \frac{\lambda \left|\Delta^{(S')}\right|}{h} k_y \, . \tag{63}$$

Figure 4(a) shows a plot of $E(k_y)$, together with the minimum-quasiparticle-excitation-
energy bounds of the S and S′ regions, for an illustrative set of parameters. With
$\Delta_c - \left|\Delta^{(S)}\right| = \left|\Delta^{(S')}\right| - \Delta_c = 0.80\,\Delta_c$ being satisfied [see Eq. (4) for the definition of the
junction's critical $s$-wave pair-potential magnitude $\Delta_c$], this system's S and S′ regions repre-
sent well-developed TSF and NSF phases, respectively. A zoom-in to the low-energy, small-$k_y$

region is shown in Fig. 4(b), including also interface-localized Andreev bound states from the spin-↑ sector.

The general form of Eq. (63) is analogous to that of the result (62), without its small $\lambda^2$-dependent correction, obtained when solving the I-TSF-edge characteristic equation (36). Thus, to leading order in small spin-orbit-coupling magnitude, the same type of linear-in-$k_y$ dispersion indicative of Majorana excitations in a chiral-$p$-wave superfluid emerges for both the I-TSF edge mode and the spin-↓ TSF-NSF interface mode. However, these modes exhibit crucial differences. Firstly, the opposite overall signs of their dispersions indicate that the propagation direction of the SS′-interface edge state is opposite to that of the IS edge state, as illustrated by Fig. 1. Furthermore, the TSF edge state is associated with quasiparticles having dominant spin-↑, i.e., majority-spin character, while the TSF-NSF interface edge state is formed by minority-spin (spin-↓) quasiparticles. Finally, the familiar form of the Majorana-edge-mode velocity for a chiral-$p$-wave superfluid that is given by the $p$-wave pair-potential magnitude divided by the Fermi wave vector applies in both cases. However, because of the way that $\Delta_\sigma^{(R)}$ depends also on $k_{\mathrm{F}\sigma}^{(R)}$ [see Eq. (19)], the Majorana-mode velocity at the SS′ interface turns out to be a measure of the $s$-wave pair-potential magnitude in the S′ part, while the velocity of the IS-edge Majorana mode depends on the $s$-wave pair-potential magnitude of the S region. Within our particular scenario where the relation $\left|\Delta^{(S)}\right| < \left|\Delta^{(S')}\right|$ holds [inferred from Eq. (3); see also Fig. 1], the SS′ interface edge mode will be faster than the IS edge mode. Thus, while the TSF region depicted in Fig. 1 has the required edge modes at its boundaries with nontopological vacuum (I) and a nontopological superfluid (NSF), the edge-mode properties are not necessarily determined solely by TSF-region parameters.

The excitation spectrum of the chiral-$p$-wave Josephson junction formed by the spin-↑ degrees of freedom from the TSF and NSF regions turns out to be unaffected by coupling to the spin-↓ sector as long as the energy of the Majorana edge mode realized in the spin-↓ sector is well-separated from that of the spin-↑ bound state. (See Sec. 2.3.3 for a detailed discussion.) With this condition, majority-spin Andreev-bound-state energies are solutions of the characteristic equation (49), yielding the explicit expression

$$E_{\uparrow\eta}(k_y) = \eta \, \frac{\lambda \, k_{\mathrm{F}\uparrow}^{(SS')}}{h} \, \sqrt{\left|\Delta^{(S)}\right|\left|\Delta^{(S')}\right|} \, \frac{\cos\left[\left(\varphi^{(S')} - \varphi^{(S)}\right)/2\right]}{\sqrt{1 + \frac{G}{\sin^2\left[\left(\varphi^{(S')} - \varphi^{(S)}\right)/2\right]}}} \,. \tag{64}$$

In Eq. (64), $\eta \in \{+, -\}$ distinguishes the positive-energy and negative-energy branches, $G$ is defined in Eq. (A.13d), and $\arccos\left(\left|\Delta^{(S)}\right|/\left|\Delta^{(S')}\right|\right) < \varphi^{(S')} - \varphi^{(S)} < 2\pi - \arccos\left(\left|\Delta^{(S)}\right|/\left|\Delta^{(S')}\right|\right)$ is required for a bound state to exist. Thus the difference in order-parameter magnitudes on opposite sides of the interface manifests in $G \neq 0$ and the restriction on the range of order-parameter phase differences across the junction. The $k_y$-independence of (64) will be modified at larger $k_y$ by corrections $\mathcal{O}(k_y^2)$ that we have neglected within the Andreev approximation. Figure 4(b) plots $E_{\uparrow\eta}(k_y)$ from Eq. (64) for a particular realization of the TSF-NSF interface together with the dispersion for the spin-↓-sector Majorana mode [$E_\downarrow(k_y)$ from Eq. (63)].

# 5 Conclusions

We have studied theoretically the subgap excitations emerging at boundaries and interfaces of the polarized 2D Fermi superfluid subject to spin-orbit coupling and $s$-wave attraction. While the specific form of 2D-Rashba-type [52–54] spin-orbit coupling has been assumed in all of our calculations, our conclusions apply more generally also to other **k**-linear types such as 2D-Dirac [68] and 2D-Dresselhaus [69, 70]. We juxtapose the properties of edge excitations at the vacuum boundary with those of Andreev bound states localized at the interface between

topological and nontopological phases of the polarized spin-orbit-coupled 2D Fermi superfluid. See Fig. 1 for a schematic overview of our system of interest.

The theoretical formalism employed for deriving the Andreev-bound-state energy dispersions and Nambu-spinor forms utilizes and extends the accurate effective description of mixed-spin quasiparticle excitations in terms of their dominant spin-projected Nambu-spinor amplitudes [30]. Section 2 provides an introduction to this formalism that is tailored to this work's objectives and contains relevant mathematical details of its extension. Our approach provides a systematic platform for mapping the low-energy physics of the spin-orbit-coupled polarized 2D Fermi gas with $s$-wave pairing to the excitation spectra of effective chiral-$p$-wave superfluids realized within majority-spin (spin-↑) and minority-spin (spin-↓) subspaces. (See Fig. 2 for an illustration.) It is a crucial feature of our theory that the intrinsic coupling between opposite-spin sectors is carefully treated to ensure realistic physical predictions. The methods developed here enable a versatile treatment of Bogoliubov quasiparticles in superfluids with multiple coupled degrees of freedom.

While the edge excitations of chiral-$p$-wave superfluids [23–29] and noncentrosymmetric superconductors [33, 71] have been studied extensively, much less attention has been focused on the interface between topological and nontopological superfluids. Our present work sheds new light on the Andreev-bound-state properties for both these venues: the insulator-superfluid (IS) boundary and the superfluid-superfluid (SS′) junction. In particular, we demonstrate how the boundary of the 2D $s$-wave Fermi superfluid in the nontopological phase establishes a Josephson junction between the chiral-$p$-wave superfluids realized in the system's spin-↑ and spin-↓ sectors, with the spin-orbit coupling determining the effective junction transparency. This is epitomized by the analogous formal structure of Eqs. (35b) and (60). We also elucidate corrections to the Majorana-edge-mode dispersion in the topological phase [Eq. (62)] arising from the coupling between opposite-spin sectors. Most importantly, we establish the emergence of a Majorana mode at the interface between topological and nontopological phases of the spin-orbit-coupled polarized 2D Fermi superfluid and elucidate its distinct physical properties. In particular, its velocity depends on the $s$-wave pair potential of the nontopological part of the system and is thus different from that of the Majorana edge mode of the topological phase with vacuum. [Compare Eqs. (62) and (63).] Having different Majorana-mode velocities at the IS boundary and the SS′ interface constitutes an experimental signature for the different origin of the Majorana excitation in both locations. Another characteristic that is different between these two Majorana modes is their opposite spin, as illustrated in Fig. 1.

Tunneling spectroscopy has been proposed as an experimental tool for identifying Bogoliubov-quasiparticle excitations with Majorana character realized in semiconductor-superconductor hybrid structures [7] by their vanishing excitation energy [9] or the equal magnitude of their particle and hole admixtures [72]. Spectroscopic techniques for measuring the energies and wave functions of quasiparticle excitations in cold-atom systems are also available [66]. Specifically, spatially resolved rf spectroscopy [73] can be adapted to measure the local density of states, thus serving as a cold-atom analog of tunneling spectroscopy [74]. The utility of this technique for probing Majorana edge modes in a 2D topological Fermi superfluid has already been pointed out [36], and we expect it to be similarly useful for facilitating experimental detection of the Majorana excitation at an interface between coexisting topological and nontopological superfluids. Spin-resolved measurements [66] would be able to confirm our prediction that the interface-localized Majorana mode has minority-spin character, in contrast to the Majorana excitation from the majority-spin sector present at the vacuum boundary.

Current interest in the spin-orbit-coupled polarized 2D Fermi superfluid has been fuelled largely by the system's topological phase simulating features of a chiral-$p$-wave superfluid. The microscopic basis for such behavior is a good separation between majority-spin and minority-

spin excitations. We have used our theoretical formalism for a systematic exploration of the effect that any residual coupling between the opposite-spin subsystems has on Andreev bound states at the IS vacuum boundary or an SS′ interface. At the SS′ interface between topological and nontopological phases, the emergent Majorana mode in the minority-spin sector and the chiral-$p$-wave junction established in the majority-spin sector turn out to be robust as long as the Andreev-bound-state energies in both subsystems are well-separated. Future work may explore the interplay between these two in the strong-coupling regime. It would also be useful to study the potential for hybridization between the Majorana modes localized at the IS boundary and SS′ interface when the S-region width $L$ becomes small (see Fig. 1). The fact that these two modes are from opposite-spin sectors leads us to expect that hybridization is even more strongly suppressed than between Majorana modes at opposite vacuum boundaries of a topological superfluid (i.e., an ISI system).

In closing, we briefly address certain features of real physical systems that have not been accounted for in our formalism. *(i) Finite-temperature ($T > 0$) effects.* The fact that the Andreev bound states considered in this work have subgap energies provides them with some protection against thermal fluctuations, as long as $k_{\mathrm{B}}T$ is smaller than their energy separation from the nearest quasiparticle continuum. This criterion limits the $k_y$ range of any dispersive mode, but it is least restrictive for chiral Majorana excitations as the latter disperse linearly around a zero-energy state. The thermal energy will also influence the size of the region indicated by hatched circles in Fig. 4(b) where a more sophisticated theoretical description of the TSF-NSF interface bound states is needed. More broadly, proper $T$-dependent values will have to be used for all parameters entering quantitative predictions presented in this work. In this context, we expect the thermal suppression of pair-potential magnitudes to result in the most significant adjustments. *(ii) Effect of a trapping potential.* Especially in cold-atom realizations of a TSF-NSF hybrid system, smoothly varying external potentials introduce spatial inhomogeneity. Our results apply most directly to situations with optical-box potentials [75], 2D versions of which have been used for trapping fermionic atoms [76] and realising an ideal Josephson junction [77]. In the more common case of harmonically trapped 2D fermion systems (see, e.g., Ref. [78]), the particle density, chemical potential and $s$-wave pair potential acquire smooth spatial variations on the scale of the trap's harmonic-oscillator length. As the Andreev bound states are localized at a system boundary or the TSF-NSF interface within much shorter length scales, their qualitative features can be expected to be largely unaffected by the trap. Quantitative predictions made within this work should also still apply if local values of, e.g., the $s$-wave pair potential are used. *(iii) Dipolar interactions* are expected to stabilize chiral-$p$-wave pairing [79–81]. Current interest in ultracold dipolar gases [82–84] motivates extension of our study to describe Andreev bound states at boundaries of such systems.

# Acknowledgements

We thank Philip Brydon for helpful discussions.

**Funding information**   This work was partially supported by the Marsden Fund of New Zealand (contract nos. VUW1713 and MAU2007) from government funding managed by the Royal Society Te Apārangi.

# A  Andreev bound states at chiral-$p$-wave-superfluid junctions

If the residual weak coupling between opposite-spin degrees of freedom in a spin-orbit-coupled polarized 2D Fermi superfluid with $s$-wave attraction is neglected, then the system is effectively split into separate spin-↑ and spin-↓ parts that constitute 2D chiral-$p$-wave superfluids with opposite chirality. A spin-conserving implementation of the SS′ interface then establishes separate junctions for the spin-↑ and spin-↓ sectors. In this Appendix, we discuss the general properties of spinless chiral-$p$-wave superfluid junctions to inform the understanding of the TSF-NSF hybrid system considered in the main part of the Article in the limit where opposite-spin subsystems are completely decoupled.

We consider an SS′ hybrid system consisting of two spinless 2D chiral-$p$-wave superfluids occupying the half-spaces $x > 0$ and $x < 0$, respectively. Within a given subspace $R \in \{S, S'\}$, the superfluid is described by a $2 \times 2$ BdG equation

$$
\begin{pmatrix}
\frac{\hbar^2}{2m^{(R)}} \hat{\mathbf{k}}^2 + v^{(R)} - \mu & \mathbf{\Delta}^{(R)} \cdot \frac{\hat{\mathbf{k}}}{k_{\mathrm{F}}^{(R)}} \\
\left[\mathbf{\Delta}^{(R)}\right]^* \cdot \frac{\hat{\mathbf{k}}}{k_{\mathrm{F}}^{(R)}} & -\left[\frac{\hbar^2}{2m^{(R)}} \hat{\mathbf{k}}^2 + v^{(R)} - \mu\right]
\end{pmatrix}
\begin{pmatrix} u^{(R)}(\mathbf{r}) \\ v^{(R)}(\mathbf{r}) \end{pmatrix}
= E \begin{pmatrix} u^{(R)}(\mathbf{r}) \\ v^{(R)}(\mathbf{r}) \end{pmatrix},
\tag{A.1}
$$

with $k_{\mathrm{F}}^{(R)} = \sqrt{2m^{(R)}\left(\mu - v^{(R)}\right)/\hbar^2}$. All system parameters, including the band-bottom shift $v^{(R)}$ and the chiral-$p$-wave order-parameter vector $\mathbf{\Delta}^{(R)} \equiv \left(\Delta_x^{(R)}, \Delta_y^{(R)}\right) = \left(\Delta^{(R)}, \gamma^{(R)} i \Delta^{(R)}\right)$, are constants within region $R$, and $\gamma^{(R)} = +1$ ($-1$) encodes the positive (negative) chirality. Translational invariance along the junction motivates the separation *Ansatz*

$$
\begin{pmatrix} u^{(R)}(\mathbf{r}) \\ v^{(R)}(\mathbf{r}) \end{pmatrix} = \begin{pmatrix} u(x) \\ v(x) \end{pmatrix} \mathrm{e}^{i k_y y},
\tag{A.2}
$$

which transforms the 2D BdG equation (A.1) into an effective 1D BdG equation for motion perpendicular to the interface,

$$
\begin{pmatrix}
\frac{\hbar^2}{2} \hat{k}_x \frac{1}{m(x)} \hat{k}_x + \frac{\hbar^2 k_y^2}{2m(x)} + v(x) - \mu & \frac{1}{k_{\mathrm{F}}(x)}\{\Delta_x(x), \hat{k}_x\} + \Delta_y(x) \frac{k_y}{k_{\mathrm{F}}(x)} \\
\frac{1}{k_{\mathrm{F}}(x)}\{\Delta_x^*(x), \hat{k}_x\} + \Delta_y^*(x) \frac{k_y}{k_{\mathrm{F}}(x)} & -\left[\frac{\hbar^2}{2} \hat{k}_x \frac{1}{m(x)} \hat{k}_x + \frac{\hbar^2 k_y^2}{2m(x)} + v(x) - \mu\right]
\end{pmatrix}
\begin{pmatrix} u(x) \\ v(x) \end{pmatrix} = E \begin{pmatrix} u(x) \\ v(x) \end{pmatrix}.
\tag{A.3}
$$

Here $\{A, B\} \equiv (AB + BA)/2$ denotes the symmetrized product of two operators, and we have allowed for all relevant parameters to be piecewise-constant;

$$
m(x) = m^{(S)} \Theta(-x) + m^{(S')} \Theta(x),
\tag{A.4a}
$$

$$
v(x) = v^{(S)} \Theta(-x) + v^{(S')} \Theta(x),
\tag{A.4b}
$$

$$
k_{\mathrm{F}}(x) = k_{\mathrm{F}}^{(S)} \Theta(-x) + k_{\mathrm{F}}^{(S')} \Theta(x),
\tag{A.4c}
$$

$$
\Delta_j(x) = \Delta_j^{(S)} \Theta(-x) + \Delta_j^{(S')} \Theta(x).
\tag{A.4d}
$$

Assuming $\mu - v^{(R)} > 0$, the general *Ansatz* for the $x$-dependent part of the Nambu spinor is

$$
\begin{pmatrix} u(x) \\ v(x) \end{pmatrix} = \left[ a_{+-}^{(S)} \begin{pmatrix} u_{+-}^{(S)} \\ v_{+-}^{(S)} \end{pmatrix} \mathrm{e}^{i k_{+-}^{(S)} x} + a_{-+}^{(S)} \begin{pmatrix} u_{-+}^{(S)} \\ v_{-+}^{(S)} \end{pmatrix} \mathrm{e}^{i k_{-+}^{(S)} x} \right] \Theta(-x)
$$

$$
+ \left[ a_{++}^{(S')} \begin{pmatrix} u_{++}^{(S')} \\ v_{++}^{(S')} \end{pmatrix} \mathrm{e}^{i k_{++}^{(S')} x} + a_{--}^{(S')} \begin{pmatrix} u_{--}^{(S')} \\ v_{--}^{(S')} \end{pmatrix} \mathrm{e}^{i k_{--}^{(S')} x} \right] \Theta(x),
\tag{A.5}
$$

with the wave numbers and Nambu-spinor entries given explicitly as

$$k_{\tau\alpha}^{(R)} = \alpha \sqrt{\left(k_F^{(R)}\right)^2 - k_y^2 - 2\left(\frac{m^{(R)}\left|\Delta^{(R)}\right|}{\hbar^2 k_F^{(R)}}\right)^2 + \tau \frac{2m^{(R)}}{\hbar^2}\sqrt{E^2 - \left|\Delta^{(R)}\right|^2\left[1 - \left(\frac{m^{(R)}\left|\Delta^{(R)}\right|}{\hbar^2\left(k_F^{(R)}\right)^2}\right)^2\right]}},$$

(A.6a)

$$u_{\tau\alpha}^{(R)} = \exp\left(i\varphi^{(R)}\right)\frac{k_{\tau\alpha}^{(R)} + \gamma^{(R)}i k_y}{\sqrt{\left(k_{\tau\alpha}^{(R)}\right)^2 + k_y^2}}$$

$$\times \sqrt{\frac{1}{2E}\left(E - \frac{m^{(R)}\left|\Delta^{(R)}\right|^2}{\hbar^2\left(k_F^{(R)}\right)^2} + \tau\sqrt{E^2 - \left|\Delta^{(R)}\right|^2\left[1 - \left(\frac{m^{(R)}\left|\Delta^{(R)}\right|}{\hbar^2\left(k_F^{(R)}\right)^2}\right)^2\right]}\right)},$$

(A.6b)

$$v_{\tau\alpha}^{(R)} = \operatorname{sgn}(E)\sqrt{\frac{1}{2E}\left(E + \frac{m^{(R)}\left|\Delta^{(R)}\right|^2}{\hbar^2\left(k_F^{(R)}\right)^2} - \tau\sqrt{E^2 - \left|\Delta^{(R)}\right|^2\left[1 - \left(\frac{m^{(R)}\left|\Delta^{(R)}\right|}{\hbar^2\left(k_F^{(R)}\right)^2}\right)^2\right]}\right)}.$$

(A.6c)

The matching conditions at the SS′ interface read

$$\begin{pmatrix} u(x) \\ v(x) \end{pmatrix}_{x\to 0^-} - \begin{pmatrix} u(x) \\ v(x) \end{pmatrix}_{x\to 0^+} = 0,$$

(A.7a)

$$\left[\frac{1}{m(x)}\frac{d}{dx}\begin{pmatrix} u(x) \\ v(x) \end{pmatrix}\right]_{x\to 0^-} - \left[\frac{1}{m(x)}\frac{d}{dx}\begin{pmatrix} u(x) \\ v(x) \end{pmatrix}\right]_{x\to 0^+} = \begin{pmatrix} 0 & -i\kappa \\ i\kappa^* & 0 \end{pmatrix}\begin{pmatrix} u(0) \\ v(0) \end{pmatrix},$$

(A.7b)

with the generally complex quantity

$$\kappa = \frac{\left|\Delta^{(S)}\right|}{\hbar^2 k_F^{(S)}}\exp\left(i\varphi^{(S)}\right) - \frac{\left|\Delta^{(S')}\right|}{\hbar^2 k_F^{(S')}}\exp\left(i\varphi^{(S')}\right).$$

(A.7c)

The $\kappa$-dependent terms on the right-hand side of Eq. (A.7b) have been neglected in previous works [24,43–46] as their formalism employed the Andreev approximation [59] from the start. Applying the conditions (A.7) to the *Ansatz* (A.5) yields the characteristic equation

$$0 = \left[\left(\frac{k_{++}^{(S')}}{m^{(S')}} - \frac{k_{+-}^{(S)}}{m^{(S)}}\right)\left(\frac{k_{-+}^{(S)}}{m^{(S)}} - \frac{k_{--}^{(S')}}{m^{(S')}}\right) - |\kappa|^2\right]\left(u_{++}^{(S')}u_{+-}^{(S)}v_{-+}^{(S)}v_{--}^{(S')} + v_{++}^{(S')}v_{+-}^{(S)}u_{-+}^{(S)}u_{--}^{(S')}\right)$$

$$+ \left[\left(\frac{k_{++}^{(S')}}{m^{(S')}} - \frac{k_{-+}^{(S)}}{m^{(S)}}\right)\left(\frac{k_{--}^{(S')}}{m^{(S')}} - \frac{k_{+-}^{(S)}}{m^{(S)}}\right) + |\kappa|^2\right]\left(u_{+-}^{(S)}v_{++}^{(S')}v_{-+}^{(S)}u_{--}^{(S')} + v_{+-}^{(S)}u_{++}^{(S')}u_{-+}^{(S)}v_{--}^{(S')}\right)$$

$$- \left(\frac{k_{++}^{(S')}}{m^{(S')}} - \frac{k_{--}^{(S')}}{m^{(S')}}\right)\left(\frac{k_{-+}^{(S)}}{m^{(S)}} - \frac{k_{+-}^{(S)}}{m^{(S)}}\right)\left(u_{++}^{(S')}v_{+-}^{(S)}v_{-+}^{(S)}u_{--}^{(S')} + v_{++}^{(S')}u_{+-}^{(S)}u_{-+}^{(S)}v_{--}^{(S')}\right)$$

$$- \left(\frac{k_{++}^{(S')}}{m^{(S')}} - \frac{k_{--}^{(S')}}{m^{(S')}}\right)\left(u_{+-}^{(S)}v_{-+}^{(S)} - v_{+-}^{(S)}u_{-+}^{(S)}\right)\left(\kappa^* u_{++}^{(S')}u_{--}^{(S')} + \kappa\, v_{++}^{(S')}v_{--}^{(S')}\right)$$

$$- \left(\frac{k_{-+}^{(S)}}{m^{(S)}} - \frac{k_{+-}^{(S)}}{m^{(S)}}\right)\left(u_{++}^{(S')}v_{--}^{(S')} - v_{++}^{(S')}u_{--}^{(S')}\right)\left(\kappa^* u_{+-}^{(S)}u_{-+}^{(S)} + \kappa\, v_{+-}^{(S)}v_{-+}^{(S)}\right),$$

(A.8)

for the interface-localized Andreev-bound-state energies.

Equation (A.8) applies generally for all situations where $\mu - \nu^{(S,S')} > 0$. Leaving a more complete analysis of the solutions to future work, we focus on the case when the Andreev approximation [59] is applicable. Within this approach, terms $\propto \kappa$ in (A.8) can be neglected, and

$$k_{\tau\alpha}^{(R)} \approx \alpha\, k_{\mathrm{F}}^{(R)} + \alpha\tau\, \frac{m^{(R)}}{\hbar^2 k_{\mathrm{F}}^{(R)}} \sqrt{E^2 - \left(\Delta^{(R)}\right)^2}, \tag{A.9a}$$

$$u_{\tau\alpha}^{(R)} \approx \alpha\, \exp\left\{ i\left[ \varphi^{(R)} + \frac{\tau\,\mathrm{sgn}(E)}{2}\,\theta^{(R)} + \alpha\,\gamma^{(R)}\vartheta_{k_y}^{(R)} \right] \right\} \sqrt{\frac{\left|\Delta^{(R)}\right|}{2|E|}}, \tag{A.9b}$$

$$v_{\tau\alpha}^{(R)} \approx \mathrm{sgn}(E)\, \exp\left[ -i\frac{\tau\,\mathrm{sgn}(E)}{2}\,\theta^{(R)} \right] \sqrt{\frac{\left|\Delta^{(R)}\right|}{2|E|}}, \tag{A.9c}$$

with the phase angle $\theta^{(R)} = \arccos\left(|E|/\left|\Delta^{(R)}\right|\right)$, and $\vartheta_{k_y}^{(R)} = \arcsin\left(k_y/k_{\mathrm{F}}^{(R)}\right)$ is reminiscent of a quasiparticle's angle of incidence on the interface. Using the approximations from Eqs. (A.9) in the characteristic equation (A.8), the latter simplifies to

$$\cos\left[ \theta^{(S')} + \theta^{(S)} + \mathrm{sgn}(E)\left( \gamma^{(S')}\vartheta_{k_y}^{(S')} - \gamma^{(S)}\vartheta_{k_y}^{(S)} \right) \right]$$
$$+ \frac{Z^2}{1+Z^2}\cos\left[ \theta^{(S')} - \theta^{(S)} + \mathrm{sgn}(E)\left( \gamma^{(S')}\vartheta_{k_y}^{(S')} + \gamma^{(S)}\vartheta_{k_y}^{(S)} \right) \right] = \frac{1}{1+Z^2}\cos\left( \varphi^{(S')} - \varphi^{(S)} \right). \tag{A.10}$$

Equation (A.10) generalizes previously obtained [43–46] forms of the characteristic equation for Andreev bound states at chiral-$p$-wave junctions to the situation where $\theta^{(S)} \neq \theta^{(S')}$, and it is also a special case of characteristic equations derived for bound states at general unconventional-superconductor junctions [47, 48]. That the interface-transparency parameter [85]

$$Z = \frac{1}{2}\left| \sqrt{\frac{m^{(S)} k_{\mathrm{F}}^{(S')}}{m^{(S')} k_{\mathrm{F}}^{(S)}}} - \sqrt{\frac{m^{(S')} k_{\mathrm{F}}^{(S)}}{m^{(S)} k_{\mathrm{F}}^{(S')}}} \right| \tag{A.11}$$

enters via the combinations $T = 1/(1+Z^2)$ and $R \equiv 1 - T = Z^2/(1+Z^2)$ that correspond, respectively, to the quantum probabilities for single-particle transmission and reflection through the interface is a well-known feature of Josephson junctions [86–88]. Application of addition theorems for trigonometric functions transforms the characteristic equation (A.10) into the form given in Eq. (60).

Analytical solution of (A.10) for the Andreev-bound-state energy $E_\alpha(k_y)$ is possible in certain limits. For basic-illustration purposes, we provide here the result for $k_y = 0$;

$$E_\eta(k_y = 0) = \eta\, E_0 \left\{ \frac{\cos^2(\varphi/2)}{1+Z^2} + \frac{1}{1+Z^2}\,\frac{F(\varphi)}{2Z^2}\left[ \sqrt{1 - 4Z^2\cos^2(\varphi/2)\frac{G}{[F(\varphi)]^2}} - 1 \right] \right\}^{\frac{1}{2}}. \tag{A.12}$$

Here $\eta = \pm$ distinguishes two branches of bound-state energies, and the parameters entering

the expression (A.12) are

$$E_0 = \sqrt{\left|\Delta^{(S)}\right|\left|\Delta^{(S')}\right|}, \tag{A.13a}$$

$$\varphi = \varphi^{(S')} - \varphi^{(S)}, \tag{A.13b}$$

$$F(\varphi) = G + Z^2 + \sin^2(\varphi/2), \tag{A.13c}$$

$$G = \frac{1}{4}\left(\sqrt{\frac{\left|\Delta^{(S')}\right|}{\left|\Delta^{(S)}\right|}} - \sqrt{\frac{\left|\Delta^{(S)}\right|}{\left|\Delta^{(S')}\right|}}\right)^2, \tag{A.13d}$$

with $Z$ given in Eq. (A.11). When $(1+2Z^2)\left|\Delta^{(S)}\right| < \left|\Delta^{(S')}\right|$, bound states exist only for values of the phase difference $\varphi$ across the junction that are within the restricted range $\varphi_0 < \varphi < 2\pi - \varphi_0$, with

$$\varphi_0 = \arccos\left[(1+2Z^2)\frac{\left|\Delta^{(S)}\right|}{\left|\Delta^{(S')}\right|}\right] \equiv \arccos\left[\frac{1+2Z^2}{\left(\sqrt{1+G}+\sqrt{G}\right)^2}\right]. \tag{A.14}$$

Without loss of generality, the definition (A.14) of $\varphi_0$ assumes $\left|\Delta^{(S)}\right| \le \left|\Delta^{(S')}\right|$. The fact that interface-localized Andreev bound states can exist only within a restricted range of $\varphi$ when pair-potential magnitudes are different on opposites sides of the junction is also true for the $s$-wave case, which was overlooked in previous work [49]. Specializing (A.12) to the case $\left|\Delta^{(S)}\right| = \left|\Delta^{(S')}\right| \equiv |\Delta|$, corresponding to having chiral-$p$-wave pair-potential components of equal magnitude on both sides of the junction, yields

$$E_\eta(k_y = 0) \to \frac{\eta\,|\Delta|\,|\cos(\varphi/2)|}{\sqrt{1+Z^2}}, \quad \text{for } G \to 0, \tag{A.15}$$

in agreement with previous results [43–46].

Panels (c) and (d) in Fig. 5 show plots illustrating parameter dependences of the $k_y = 0$ Andreev-bound-state energies at junctions between chiral-$p$-wave superfluids. Comparison with the results obtained for bound states localized at the interface between $s$-wave superfluids [panels (a) and (b) of the same figure] reveals distinctive properties of chiral-$p$-wave junctions. One particularly striking feature is the different effect normal reflection (signified by $Z > 0$) has on the energy spectrum [44, 45]. For a junction of $s$-wave superfluids, the crossing at $\varphi = \pi$ becomes an anticrossing, while the energies at $\varphi = \varphi_0$ and $2\pi - \varphi_0$ remain unaffected. In contrast, for a chiral-$p$-wave junction, the crossing at $\varphi = \pi$ remains unaffected, while the magnitude of $\varphi_0$ gets reduced and, eventually, the energies at $\varphi = 0$ and $2\pi$ are suppressed by normal reflection. The physical origin for this qualitatively different behavior lies in the additional wave-vector-dependent phase shifts incurred for normal and Andreev reflection off the interface with a chiral-$p$-wave superfluid, which shift the configuration with constructive interference between normal and Andreev reflection for interface-localized bound states from $\varphi = 0$ for an $s$-wave SS′ junction to $\varphi = \pi$ for the chiral-$p$-wave case.

Analytical solutions of the characteristic equation (A.10) can also be obtained for a fully transparent interface; i.e., $Z \to 0$. Assuming the velocity parallel to the interface to have a magnitude not exceeding $v_{y,\max}$ defined via $v_{y,\max}^2 = v_F^2 \sqrt{G}/(\sqrt{1+G}+\sqrt{G})$, where $v_F = \hbar k_F^{(S)}/m^{(S)}$ ($\equiv \hbar k_F^{(S')}/m^{(S')}$ for $Z = 0$), we find

$$E_\eta(k_y) \to \frac{\eta\,E_0\,\cos\left(\varphi_{k_y}^{(\eta)}/2\right)}{\sqrt{1+\frac{G}{\sin^2\left(\varphi_{k_y}^{(\eta)}/2\right)}}}, \quad \text{for } Z \to 0, \tag{A.16a}$$

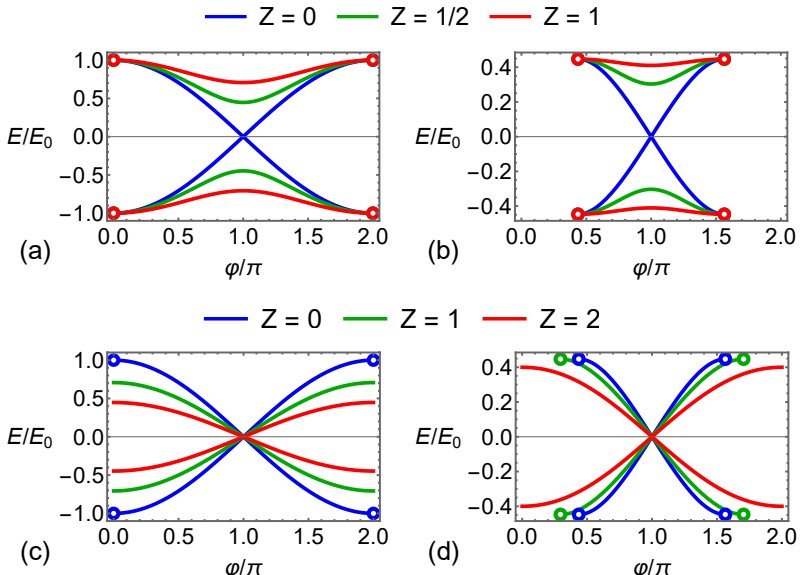

Figure 5: Energy $E$ of Andreev bound states at SS′ interfaces, obtained within the Andreev approximation. Results shown in panels (a) and (b) [(c) and (d)] are for a junction between two 2D $s$-wave superfluids with pair potentials $\left|\Delta^{(S/S')}\right|\exp\{(-/+)i\varphi/2\}$ [two 2D chiral-$p$-wave superfluids with pair potentials $\left|\Delta^{(S/S')}\right|\exp\{(-/+)i\varphi/2\}\{k_x+\gamma^{(S/S')}ik_y\}/k_F^{(S/S')}$ for the case with vanishing wave-vector component parallel to the interface, i.e., $k_y = 0$]. Situations with equal quasiparticle-gap magnitudes on both sides of the SS′ interface are depicted in panels (a) and (c), corresponding to $G = 0$. [See Eq. (A.13d) for the definition of $G$.] For comparison, panel (b) [(d)] is for $G = 4/5$. The different curves within panels (a) and (b) [(c) and (d)] are for values $Z = 0$, $1/2$, and $1$ [$Z = 0$, $1$ and $2$] of the interface-transparency parameter $Z$ defined in Eq. (A.11). Circles indicate end points of $\varphi$ ranges for which bound states exist. The energy unit $E_0$ is defined in Eq. (A.13a).

with

$$\varphi_{k_y}^{(\eta)} = \varphi + \eta\left(\gamma^{(S)}\vartheta_{k_y}^{(S)} - \gamma^{(S')}\vartheta_{k_y}^{(S')}\right), \tag{A.16b}$$

and $\varphi_0 < \varphi_{k_y}^{(\eta)} < 2\pi - \varphi_0$. Thus, $G \neq 0$ causes a linear $k_y$ dependence of the Andreev bound-state energies for the same-chirality junction, which normally (i.e., for $G = 0$) is only a feature for opposite-chirality junctions [44, 45].

The situation where the SS′ interface becomes a dividing wall and the two chiral-$p$-wave superfluids are disconnected can be described by taking the limit $Z \to \infty$ in the characteristic equation (A.10) or, equivalently, by setting $T \to 0$ in (60). The resulting characteristic equation can be expressed in the factorized form

$$\cos\left[\theta^{(S')} + \text{sgn}(E)\gamma^{(S')}\vartheta_{k_y}^{(S')}\right]\cos\left[\theta^{(S)} - \text{sgn}(E)\gamma^{(S)}\vartheta_{k_y}^{(S)}\right] = 0, \tag{A.17}$$

which is satisfied if either one of the two individual cosine factors appearing on the left-hand side of Eq. (A.17) is separately equal to zero. Solution of the two resulting conditions yield the Majorana edge modes [23–29] associated with the left boundary of the S′ region and the right boundary of the S region, having the respective dispersions

$$E^{(S')}(k_y) = \gamma^{(S')}\frac{|\Delta^{(S')}|}{k_F^{(S')}}k_y, \quad E^{(S)}(k_y) = -\gamma^{(S)}\frac{|\Delta^{(S)}|}{k_F^{(S)}}k_y. \tag{A.18}$$

# B Majorana edge modes of chiral-*p*-wave superfluids

The two independent solutions of the characteristic equation (A.17), arising in the limit where the SS′ interface is a dividing wall, describe the edge states propagating along the wall in the two separated S and S′ regions. Here we investigate the evanescent states associated with the energy dispersions $E^{(S')}(k_y)$ and $E^{(S)}(k_y)$ from Eq. (A.18).

To be specific, we start by considering the edge state of the S′ region. The $x$-dependent part of its Bogoliubov spinor is given by the $x > 0$ part of the *Ansatz* (A.5), involving the complex wave-vector components $k_{\tau\alpha}^{(S')}$ and two-spinors $\left(u_{\tau\alpha}^{(S')}, v_{\tau\alpha}^{(S')}\right)^T$ for which $\alpha\tau \in \{++, --\}$. Using the Andreev-approximation expressions given in Eqs. (A.9) and neglecting terms of magnitude $\mathcal{O}\left(\left[k_y/k_F^{(S')}\right]^2\right)$ for consistency, we find

$$\begin{pmatrix} u_{++}^{(S')} \\ v_{++}^{(S')} \end{pmatrix}_{E=E^{(S')}(k_y)} = \begin{pmatrix} u_{--}^{(S')} \\ v_{--}^{(S')} \end{pmatrix}_{E=E^{(S')}(k_y)} \propto \begin{pmatrix} e^{\frac{i}{2}\varphi^{(S')}}(1+i) \\ e^{-\frac{i}{2}\varphi^{(S')}}(1-i) \end{pmatrix}. \tag{B.1}$$

With this result, the properly normalized Bogoliubov-spinor wave function (A.2) for the edge state having wave-vector component $k_y$ parallel to the interface in the S′ region is obtained as

$$\begin{pmatrix} u_{k_y}^{(S')}(\mathbf{r}) \\ v_{k_y}^{(S')}(\mathbf{r}) \end{pmatrix} = \frac{\Theta(x)}{\sqrt{2\pi l_{coh}^{(S')}}} \sin\left(k_F^{(S')}x\right) \exp\left(-\frac{x}{l_{coh}^{(S')}}\right) \exp\left(ik_y y\right) \begin{pmatrix} e^{\frac{i}{2}\varphi^{(S')}}(1+i) \\ e^{-\frac{i}{2}\varphi^{(S')}}(1-i) \end{pmatrix}, \tag{B.2}$$

with the S′ region's superfluid coherence length scale $l_{coh}^{(S')} = \hbar^2 k_F^{(S')}/\left(m^{(S')}\left|\Delta^{(S')}\right|\right)$. Performing a similar calculation for the edge state propagating along the SS′ interface in the S region yields the corresponding Bogoliubov spinor

$$\begin{pmatrix} u_{k_y}^{(S)}(\mathbf{r}) \\ v_{k_y}^{(S)}(\mathbf{r}) \end{pmatrix} = \frac{\Theta(-x)}{\sqrt{2\pi l_{coh}^{(S)}}} \sin\left(k_F^{(S)}x\right) \exp\left(\frac{x}{l_{coh}^{(S)}}\right) \exp\left(ik_y y\right) \begin{pmatrix} e^{\frac{i}{2}\varphi^{(S)}}(1-i) \\ e^{-\frac{i}{2}\varphi^{(S)}}(1+i) \end{pmatrix}. \tag{B.3}$$

Both spinors (B.2) and (B.3) satisfy

$$\left[u_{k_y}^{(R)}(\mathbf{r})\right]^* = v_{-k_y}^{(R)}(\mathbf{r}), \tag{B.4}$$

which is the defining relation for a Majorana mode. This is so because, in our system of interest, the particle-hole conjugation operation $C$ is defined via [4]

$$C\begin{pmatrix} u \\ v \end{pmatrix} = \begin{pmatrix} v^* \\ u^* \end{pmatrix}. \tag{B.5}$$

Thus, it is possible to form a general superposition of plane-wave edge states

$$\Psi^{(R)}(\mathbf{r}, t) = \int_0^{k_F^{(R)}} \frac{dk_y}{2\pi} \left\{ f(k_y) \exp\left[-\frac{i}{\hbar}E^{(R)}(k_y)t\right] \begin{pmatrix} u_{k_y}^{(R)}(\mathbf{r}) \\ v_{k_y}^{(R)}(\mathbf{r}) \end{pmatrix} \right. \tag{B.6}$$

$$\left. + \left[f(k_y)\right]^* \exp\left[\frac{i}{\hbar}E^{(R)}(k_y)t\right] \begin{pmatrix} u_{-k_y}^{(R)}(\mathbf{r}) \\ v_{-k_y}^{(R)}(\mathbf{r}) \end{pmatrix} \right\},$$

that satisfies $C\Psi^{(R)}(\mathbf{r}, t) = \Psi^{(R)}(\mathbf{r}, t)$ and, therefore, describes a particle that is its own particle-hole conjugate (i.e., *antiparticle*).

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
