# Peer review of "Andreev bound states at boundaries of polarized 2D Fermi superfluids with s-wave pairing and spin-orbit coupling"

_SciPost Physics, doi:SciPost Phys. 14, 115 (2023)_

## Round 1 · Referee Report · Anonymous (Referee 1) · 2022-11-28

Strengths

  1. Interesting proposal to look at edge states between coexisting phases that arise from a first-order topological phase transition
  2. Relatively simple analytic results for the Andreev bound states

Weaknesses

  • The predicted edge states might be difficult to realize and probe in cold-atom experiments

Report

This paper considers the spin-orbit-coupled polarized 2D Fermi gas and investigates the low-energy excitations that can exist at the interface between two coexisting phases with differing topology. In particular, the authors consider the boundary between a topological superfluid and a nontopological one, which could be realized at a first-order topological transition where there is spatial coexistence between different superfluids. Using the Bogoliubov–de Gennes formalism, they derive analytic expressions for the case where the spin-orbit coupling strength is smaller than the Zeeman splitting and they show that it is important to include the coupling between opposite-spin sectors (which is naturally present in this setup). They show that there can be a robust Majorana mode at the interface between topological and nontopological superfluids, provided that the Andreev bound states are well separated in energy.

Overall, this is a well-written paper and it provides a solid contribution to the field. It is quite technical in parts, which is appropriate for the aim of the paper, but it would have helped to have a clear summary of the results in the abstract, i.e., beyond vague statements like “the results extend current knowledge….”. I would have also liked to see more discussion of how the predicted states might be realized and probed in cold atom experiments. My specific comments are below:

  1. What is the effect of temperature on the results? In particular, how robust are the edge states to thermal fluctuations? This is pertinent to cold-atom experiments where it is often challenging to access low temperatures, especially when applying lasers to simulate spin-orbit coupling.

  2. Are the results sensitive to any underlying trapping potential, e.g., a harmonic trap?

  3. It was not clear to me how these states might be probed in practice. The authors refer to a proposal in Ref [69] which is based on a tunnelling measurement, but it would be good see some more details of how this might be adapted to the cold-atom case.

  4. I would be curious to know how the results might generalize to other types of interactions, e.g., dipolar interactions. Could this be used to further enhance p-wave superfluidity?

Requested changes

  • Modify abstract to include concrete results, e.g., the conditions for the existence of a robust Majorana mode.

  • Address comments above

---

## Round 1 · Referee Report · Anonymous (Referee 2) · 2022-12-26

Report

Within the spin-resolved Bogoliubov–de Gennes formalism, the authors studied the interface properties of superfluids consisting of 2D Fermi gases subject to s-wave pairings, very small spin-orbit couplings and large Zeeman fields. They considered the subgap states in two systems: the edge sates of topological or nontopological superfluid and the bound states at the interface between topological and nontopological superfluids. The paper is well written with detailed derivations, simple analytical expressions and careful illustrations. I have only one concern that the abstract is unclear in delivering the message. The authors may provide an introduction to the theoretical formalism and the implications of the subgap states. I would like to recommend its publication after improving the abstract.

---

## Round 2 · Author Response

We thank both referees for their constructive reviews of our manuscript; implementing the useful suggestions made in the reports has definitely improved our manuscript. Below we elaborate in detail on the specific changes that have been made in response to the referees’ reports.

Response to Report 1 and associated changes:

We are grateful to the referee for their thoughtful report and positive evaluation of our work. In reply, we now quote each of the specific suggestions from the report and describe their implementation in the revised manuscript.

  1. “Modify abstract to include concrete results, e.g., the conditions for the existence of a robust Majorana mode.”

This suggestion chimes with a similar recommendation made by the other referee. For the revised version of our manuscript, we have completely rewritten the abstract to deliver a clearer and more complete overview of our methods and results.

  1. “What is the effect of temperature on the results? In particular, how robust are the edge states to thermal fluctuations? This is pertinent to cold-atom experiments where it is often challenging to access low temperatures, especially when applying lasers to simulate spin-orbit coupling.”

We have added the part ‘(i) Finite-temperature (T > 0) effects … significant adjustments.’ towards the end of the Conclusion section to discuss this important point.

  1. “Are the results sensitive to any underlying trapping potential, e.g., a harmonic trap?”

This question is addressed by the newly added sentences ‘(ii) Effect of a trapping potential … pair potential are used.’ in the last paragraph of the Conclusions section, which also contain citations to new Refs. [75-78].

  1. “It was not clear to me how these states might be probed in practice. The authors refer to a proposal in Ref [69] which is based on a tunnelling measurement, but it would be good (to) see some more details of how this might be adapted to the cold-atom case.”

We have expanded the discussion of experimental probes, which now fills an entire paragraph in the Conclusions section (‘Tunneling spectroscopy … vacuum boundary.’) and includes citations to new Refs. [73] and [74].

  1. “I would be curious to know how the results might generalize to other types of interactions, e.g., dipolar interactions. Could this be used to further enhance p-wave superfluidity?

We thank the referee for this interesting question. A preliminary answer is given in the part ‘(iii) Dipolar interactions … such systems.’ at the end of the Conclusion section, where we also express the sentiment that further research in this area would be desirable. New Refs. [79-84] are cited in this context.

Response to Report 2 and associated changes:

We thank the referee for being fully supportive of our work. Quoting from the report:

“I have only one concern that the abstract is unclear in delivering the message. The authors may provide an introduction to the theoretical formalism and the implications of the subgap states. I would like to recommend its publication after improving the abstract.”

To address the referee’s concern, we have completely rewritten the abstract so that it now contains the type of detailed information requested in the above quote from the report.

Additional changes made in the resubmitted manuscript:

In addition to changes made in response to the referee reports, we have made minor revisions to describe more precisely the connections between results presented in our manuscript and earlier work on Andreev bound states in unconventional superconductors. In particular,

(a) we reformulated footnote 1, adding the part ‘is implicit in … s-wave superfluids’ and citing new Refs. [47] and [48],

(b) revised the sentence below Eq. (50), adding the part ‘the surface bound … [48,62], including’ with citations to new Refs. [48] and [62], and

(c) revised the paragraph below Eq. (74), adding the part ‘, and it is also a special case of characteristic equations derived for bound states at general unconventional-superconductor junctions’ [47,48]’.

---

## Round 2 · List of Changes

This is included as part of the author comments; see above.

---

## Editorial Decision

published